# Injecting Multimodal Information into Rigid Protein Docking via Bi-level Optimization

**Ruijia Wang[12]*†  Yiwu Sun[1]†  Yujie Luo[1]†  Shaochuan Li[1]†  Cheng Yang[2]**
**Xingyi Cheng[1]  Hui Li[1]‡  Chuan Shi[2]‡  Le Song[1]‡**
[1] BioMap Research
[2] Beijing University of Posts and Telecommunications
{wangruijia, yangcheng, shichuan}@bupt.edu.cn,
{yiwu, luoyujie, shaochuan, xingyi, lihui, songle}@biomap.com

## Abstract

The structure of protein-protein complexes is critical for understanding binding dynamics, biological mechanisms, and intervention strategies. Rigid protein docking, a fundamental problem in this field, aims to predict the 3D structure of complexes from their unbound states without conformational changes. In this scenario, we have access to two types of valuable information: sequence-modal information, such as coevolutionary data obtained from multiple sequence alignments, and structure-modal information, including the 3D conformations of rigid structures. However, existing docking methods typically utilize single-modal information, resulting in suboptimal predictions. In this paper, we propose xTrimoBiDock$^\alpha$ (or BiDock for short)[4], a novel rigid docking model that effectively integrates sequence- and structure-modal information through bi-level optimization. Specifically, a cross-modal transformer combines multimodal information to predict an inter-protein distance map. To achieve rigid docking, the roto-translation transformation is optimized to align the docked pose with the predicted distance map. In order to tackle this bi-level optimization problem, we unroll the gradient descent of the inner loop and further derive a better initialization for roto-translation transformation based on spectral estimation. Compared to baselines, BiDock achieves a promising result of a maximum 234% relative improvement in challenging antibody-antigen docking problem.

## 1 Introduction

Protein-protein interactions (PPIs) are essential to the basic functioning of cells and larger biological systems. Due to their importance, elucidating such interactions up to atomic detail is necessary for understanding multicomponent complexes like ribosomes and discovering protein-based drugs, e.g., antibodies and peptides. However, the experimental golden standard for determining the structure of protein complexes, such as X-ray crystallography and cryo-EM, is extremely time-consuming.

Computational protein docking [49, 7, 57, 45] provides an alternative route to predict the 3D structures of complexes from unbound states. Here, we focus on the fundamental problem of rigid protein docking [25] where no deformations occur within any protein during the docking process.

---

*Work done during an internship at BioMap

†Contributed equally to this research. Each author's contribution is provided in Section 5.

‡Corresponding authors

[4]xTrimoBiDock is a member of BioMap's large-scale AI engine "xTrimo" series. "$\alpha$" denotes the academic version, to distinguish it from the commercial product xTrimoBiDock.

37th Conference on Neural Information Processing Systems (NeurIPS 2023).

This assumption is reasonable in many biological environments and stabilizes the prediction of natural structures. Therefore, all we need is an appropriate SE(3) transformation shown in Figure 1, i.e., roto-translation transformation, that places the ligand protein at the correct orientation and location concerning the receptor protein. In this context, two types of information are available. The first is sequence-modal information, such as the coevolutionary signals captured in multiple sequence alignments (MSAs). The second is structure-modal information, like 3D coordinates of rigid bodies and bond angles. Both types of information are indispensable for performing rigid protein docking.

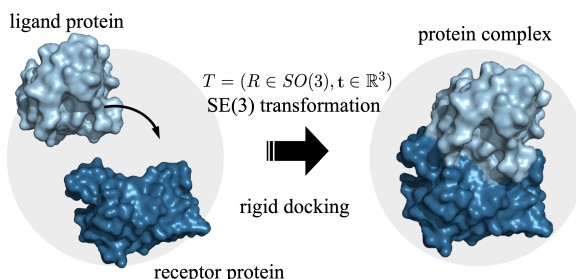

Figure 1: Surface views of rigid protein docking. Keep receptor protein at a fixed location, and a roto-translation transformation $T = (R, \mathbf{t})$ is predicted to place ligand protein at the correct docked pose.

Classical docking software [13, 19, 46, 29, 58] generally follow a three-step framework for predicting complex structures. Firstly, a large number of candidate structures are randomly sampled to explore the conformational space of the complex. Secondly, a scoring function is employed to evaluate and rank the sampled structures based on their compatibility with the binding interface. Finally, the top-ranked structures are refined using an energy model to improve their accuracy and eliminate steric clashes. Due to the evaluation of lots of candidates, these methods tend to be computationally expensive, particularly for high-throughput workflows.

Recently, deep learning has shown significant computational speed-up in this field. EquiDock [25] has emerged as a pioneering method for applying deep learning to rigid protein docking. However, it solely relies on structure-modal information, neglecting the valuable sequence-modal information in databases. This drawback hampers its ability to capture evolutionary constraints and exploit the intricate sequence-structure relationships. Building upon the success of AlphaFold2 [28], AlphaFold-Multimer [23] has been hailed as a breakthrough in directly folding complex structures from amino acid sequences, which considers the sequence-modal information but fails to effectively utilize the given rigid structures, leading to unnatural complexes in some cases.

To overcome the aforementioned limitations, we propose xTrimoBiDock (or BiDock for short), a novel rigid docking model that seamlessly integrates sequence- and structure-modal information through bi-level optimization. (i) To effectively utilize multimodal information, we introduce a cross-modal transformer that injects multimodal information into an inter-protein distance map. (ii) To satisfy rigid docking, we optimize a roto-translation transformation to minimize the difference between the docked pose of unbound proteins and the predicted inter-protein distance map. This framework naturally lends itself to a nested bi-level optimization paradigm, where the outer loop is the learning of the cross-modal transformer and the inner loop is dedicated to solving the roto-translation transformation. Inspired by gradient-based bi-level optimization [24, 32, 11], we unroll the gradient descent of the inner loop for approximation. Due to the ruggedness of the optimization landscape, a substantial number of iterations are required for convergence. Thus, we further derive a better initialization for the roto-translation transformation using spectral estimation. Extensive experiments conducted on diverse datasets and evaluation protocols validate the effectiveness of BiDock.

In summary, our contributions are three-fold:

- We effectively leverage sequence- and structure-modal information for rigid protein docking. By naturally integrating the fusion of multimodalities and the docking of rigid bodies through bi-level optimization, we set up a new avenue for solving rigid protein docking.

- We solve the above bi-level optimization with unrolled gradient and spectral initialization. By unrolling the gradient descent of the inner loop and deriving a spectral estimation for initialization, BiDock enhances the convergence and controls the computational cost.

- Comprehensive experiments on three representative datasets demonstrate the effectiveness of the proposed model. Compared to state-of-the-art baselines, BiDock achieves the maximum 234% relative improvement in challenging but practical antibody-antigen docking.

## 2  Related Work

**Molecular Docking**   Molecular docking aims to characterize the binding poses between small molecule compounds and protein targets [42, 43, 63], playing an essential role in the discovery of effective and safe treatments for various diseases [48]. Traditional computational methods have been greatly enhanced by deep learning techniques [53], which offer increased expressive power in identifying, processing, and extrapolating complex patterns in molecular data [39, 31, 21, 35, 4, 38, 16]. It is important to note that these methods have primarily focused on molecular ligands and often assume the availability of known binding pockets, limiting their direct applicability to protein-protein docking. Protein-protein docking is more challenging due to larger, flexible proteins and the need to predict unknown binding interfaces.

**Protein Docking**   Computational docking software [46, 47, 44, 57, 58, 45, 15] predict complex structures based on a framework of candidate sampling, ranking [37, 6, 22], and refinement [50]. These methods can be financially restrictive and time-consuming, as they often involve scoring and ranking thousands of candidate structures. Recently, deep learning has made significant contributions to structural biology [30, 17, 18, 36]. Notably, AlphaFold2 [28] and RoseTTAFold [2] have been employed to improve protein structure prediction from various angles [26, 40], such as integrating physics-based docking methods [29] or extending multiple sequence alignments [10]. Additionally, methods like AlphaFold-Multimer [23] and HSRN [27] have been developed to simultaneously fold and dock two proteins. Despite their remarkable achievements, these methods violate the rigidity of rigid docking and do not consider unbounded structures. EquiDock [25] is tailored for effective rigid docking but does not fully leverage the evolutionary information encoded in protein sequences, resulting in limited performance improvement compared to traditional docking software.

**Bi-level Optimization**   Bi-level optimization has gained attention in the deep learning community for its ability to handle nested problem structures. It finds applications in various domains, including hyperparameter optimization [14] and metaknowledge extraction [24]. Traditional bi-level optimization methods rely on game theory [51] or mathematical programming [9], which may not scale well to large datasets or have strict mathematical requirements. Alternatively, gradient descent methods offer a promising solution, and they can be divided into explicit gradient update [60], explicit proxy update [1, 8], implicit function [41], and closed-form methods [59]. The first three are approximation methods suitable for general functions, while the last one is an accurate method specifically designed for certain functions. Recent surveys [33, 11] provide a more comprehensive review of these methods. In addition, the merging AI4Science directions, such as topology design [61] and protein representation learning [12], also present nested problem structures that can be compatible with bi-level optimization. By incorporating bi-level optimization into rigid protein docking, we anticipate that our work will have a significant impact at the intersection of these research areas.

## 3  BiDock Methodology

**Multimodal Input**   The available information on the sequence modality mainly includes information inside the primary sequence itself and co-evolutionary information from MSAs. Following the existing work [28, 23], we extract type features $F^{typ} \in \mathbb{R}^{N_{res} \times 21}$ and primary pair features $F^{pp} \in \mathbb{R}^{N_{res} \times N_{res} \times 73}$ from the primary sequence, where $N_{res}$ is the number of residues. In terms of MSAs, we leverage cluster MSA features $F^{msa} \in \mathbb{R}^{N_{cls} \times N_{res} \times 49}$ , where $N_{cls}$ is the number of cluster centers. For the structure modality, we extract angle features $F^{ang} \in \mathbb{R}^{N_{res} \times 57}$ and pair features $F^{p} \in \mathbb{R}^{N_{res} \times N_{res} \times 88}$ from the rigid protein structures. The angle features provide the orientation and position of each amino acid, while the pair features contain the distance information between amino acids. We present a brief introduction for features and give further details in Appendix B.

**Architecture**   The illustration of the framework is presented in Figure 2. Given sequence-modal features $\{F^{typ}, F^{msa}, F^{pp}\}$ and structure-modal features $\{F^{ang}, F^{p}\}$, the cross-modal transformer with parameters $\phi$ transforms available features and integrates them to predict an inter-protein distance map $\hat{D}$. To maintain the rigid-body assumption, we define the objective function of learning roto-translation transformation $(R, \mathbf{t})$ based on the distance map $\hat{D}$ and coordinates of rigid proteins

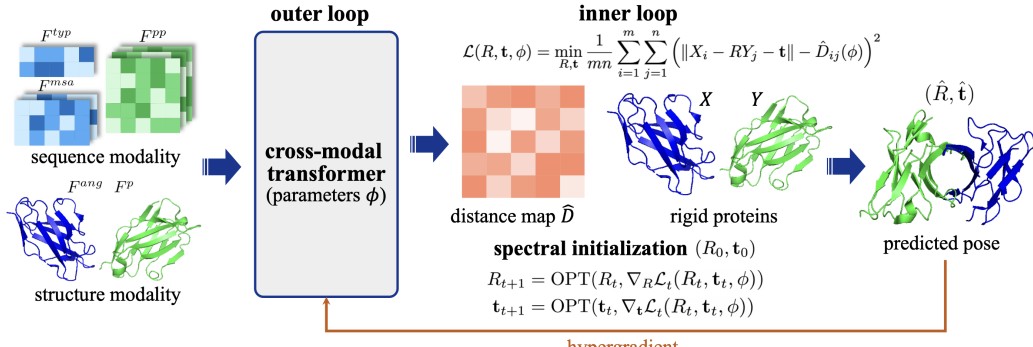

Figure 2: The overall framework of BiDock. Taking the information from sequence and structure modality as input, the cross-modal transformer fuses them and predicts an inter-protein distance map. To achieve rigid docking, the roto-translation transformation is learned by minimizing the difference between the docked pose of rigid structures and the predicted distance map. We unroll the gradient descent and further derive a spectral initialization to address the formed bi-level optimization.

$\{X \in \mathbb{R}^{3 \times m}, Y \in \mathbb{R}^{3 \times n}\}$ as follows

$$\mathcal{L}(R, \mathbf{t}, \phi) = \min_{R, \mathbf{t}} \frac{1}{mn} \sum_{i=1}^{m} \sum_{j=1}^{n} \left( \|X_i - RY_j - \mathbf{t}\| - \hat{D}_{ij}(\phi) \right)^2, \tag{1}$$

where $X_i$ is the $i$-th column of $X$ and $Y_j$ is the $j$-th column of $Y$. Considering that optimizing roto-translation transformation $(R, \mathbf{t})$ and parameters of cross-modal transformer $\phi$ constitutes a nested structure, we reformulate the rigid protein docking as a bi-level optimization problem

$$\phi^* = \operatorname*{argmin}_{\phi} \mathcal{L}^{out} \left( R^*(\phi), \mathbf{t}^*(\phi), \phi \right) \quad \text{s.t. } R^*(\phi), \mathbf{t}^*(\phi) = \operatorname*{argmin}_{R, \mathbf{t}} \mathcal{L}(R, \mathbf{t}, \phi), \tag{2}$$

where $\mathcal{L}^{out}$ refers to the outer loss for the cross-modal transformer and its specific details will be introduced in the following subsections. To solve this bi-level optimization, we unroll the gradient descent of the inner loop to approximate $(R^*(\phi), \mathbf{t}^*(\phi))$ by

$$\begin{aligned} R_{t+1} &= \operatorname{OPT}(R_t, \nabla_R \mathcal{L}_t (R_t, \mathbf{t}_t, \phi)), \\ \mathbf{t}_{t+1} &= \operatorname{OPT}(\mathbf{t}_t, \nabla_{\mathbf{t}} \mathcal{L}_t (R_t, \mathbf{t}_t, \phi)), \end{aligned} \tag{3}$$

where OPT represents the optimization algorithm, such as the stochastic gradient descent (SGD). However, due to the complexity of the problem and the rugged optimization landscape, the gradient descent in the inner loop often requires a large number of iterations to converge and faces challenges in finding the global minima. Therefore, we further derive spectral initialization to provide a more favorable starting point for better convergence.

## 3.1 Cross-modal Transformer

Based on the multimodal features, the cross-modal transformer predicts an inter-protein distance map, as depicted in Figure 3. We first use multilayer perceptrons (MLPs) to capture the nonlinear relationship inside features and project them into the same space

$$\begin{aligned} P^{pp} &= \operatorname{MLP}(F^{pp}) \quad P^p = \operatorname{MLP}(F^p) \\ M^{typ} &= \operatorname{MLP}(F^{typ}) \quad M^{msa} = \operatorname{MLP}(F^{msa}) \quad M^{ang} = \operatorname{MLP}(F^{ang}), \end{aligned} \tag{4}$$

where $\{P^{pp}, P^p \in \mathbb{R}^{N_{res} \times N_{res} \times c_z}\}$ are transformed pair features, $\{M^{typ}, M^{ang} \in \mathbb{R}^{N_{res} \times c_m}\}$ and $M^{msa} \in \mathbb{R}^{N_{cls} \times N_{res} \times c_m}$ are transformed intra-sequence features. Then we add pair features as the input pair feature $P$ of Evoformer [28]

$$P = P^{pp} + P^p. \tag{5}$$

Similarly, we integrate intra-sequence features as another input

$$M = \left[ (M^{typ} + M^{msa}) \| M^{ang} \right], \tag{6}$$

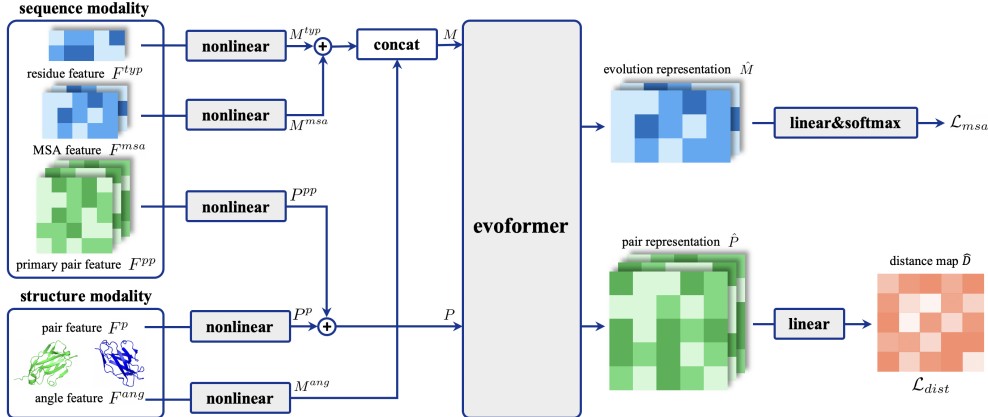

Figure 3: Details on the architecture of cross-modal transformer that enables the interaction of different modalities and outputs updated representations. In particular, the pair representation is used to predict the inter-protein distance map.

where $\|$ is the concatenation. Please note that the broadcast operation is used to make the dimensions consistent. Specifically, $M^{typ}$ is broadcasted along the newly added first dimension during addition. Similarly, broadcasting is applied to the newly added first dimension of $M^{ang}$ for concatenation. Through Evoformer, pair representation $\hat{P}$ and evolution representation $\hat{M}$ are obtained

$$\hat{P}, \hat{M} = \text{EVOFORMER}(P, M). \tag{7}$$

Finally, we utilize pair representation $\hat{P}$ to predict the inter-protein distance map. Here we define two loss functions to supervise the learning of distance map and evolution representation. Concerning the distance map, we can directly calculate the ground truth from rigid proteins. However, this ground truth is naturally noisy because of the experimental resolution. In light of successful practice in protein structure prediction [28, 23], we use a discretized distance map to replace the exact one. Specifically, distances are discretized into 64 bins ranging from 2 to 22 Å. The prediction of discretized distances is converted into a classification problem by

$$\bar{D}_{ij} = \sigma(W(\hat{P}_{ij} + \hat{P}_{ji})), \tag{8}$$

where $W$ is the learnable parameter and $\sigma$ is the activation function. Then the cross-entropy loss averaged over all residue pairs is

$$\mathcal{L}_{dist} = -\frac{1}{mn} \sum_{i,j} \sum_{k=1}^{64} \bar{G}_{ij}^k \log \bar{D}_{ij}^k, \tag{9}$$

where $\bar{G}_{ij}^k$ represents the $k$th-element of one-hot encoding of discretized actual distance. It is worth noting that the predicted distance map $\hat{D}$ can be obtained by using the mean of each bin.

Inspired by the masked language model [20], we leverage the evolution representation to reconstruct masked MSA values. We consider 23 classes, including 20 common amino acid types, an unknown type, a gap token, and a mask token, and introduce the mask policy in Appendix B. Thus, a masked MSA loss can be defined as follows

$$\bar{M} = \text{Softmax}(W\hat{M}), \tag{10}$$

$$\mathcal{L}_{msa} = -\frac{1}{N_{mask}} \sum_{i=1}^{N_{cls}} \sum_{j \in N_{mask}} \sum_{k=1}^{23} A_{ij}^k \log(\bar{M}_{ij}^k), \tag{11}$$

where $N_{mask}$ denotes the number of masked tokens and $A$ is the ground truth.

Similar to Equation (1), the SE(3) transformation optimized in the inner loop will exhibit differences from the ground truth complex, generating hypergradients through the cross-modal transformer

$$\mathcal{L}^{in}(R^*(\phi), \mathbf{t}^*(\phi)) = \frac{1}{mn} \sum_{i=1}^{m} \sum_{j=1}^{n} \left( \|X_i - R^*(\phi)Y_j - \mathbf{t}^*(\phi)\| - D_{ij} \right)^2, \tag{12}$$

where $D$ is the ground truth distance between amino acids. Overall, the outer loss for the cross-model transformer is

$$\mathcal{L}^{out} = \lambda_1 \mathcal{L}_{dist} + \lambda_2 \mathcal{L}_{msa} + \lambda_3 \mathcal{L}^{in}, \tag{13}$$

where hyperparameters $\lambda_1$, $\lambda_2$ and $\lambda_3$ balance the importance of different loss terms.

## 3.2 Unrolled Algorithm for Hypergradient

To solve the bi-level optimization formulated as Equation (2), hyper gradient $\boldsymbol{d}_\phi \mathcal{L}^{out}$ is required in the outer level and can be unrolled via the chain rule

$$\boldsymbol{d}_\phi \mathcal{L}^{out} = \frac{\partial \mathcal{L}^{out}}{\partial (R^*, \mathbf{t}^*)} \frac{\partial (R^*(\phi), \mathbf{t}^*(\phi))}{\partial \phi} + \frac{\partial \mathcal{L}^{out}}{\partial \phi}. \tag{14}$$

One method to calculate it is approximating $(R^*(\phi), \mathbf{t}^*(\phi))$ via optimizer

$$(R_t, \mathbf{t}_t) = \text{OPT}\left(R_{t-1}, \mathbf{t}_{t-1}, \phi\right), \quad t = 1, \cdots, T, \tag{15}$$

where $T$ denotes the number of iterations. We explicitly calculate the gradients

$$\tilde{D}_{ij} = X_i - R_t Y_j - \mathbf{t}_t,$$

$$\nabla_\mathbf{q} \mathcal{L}_t\left(R_t, \mathbf{t}_t, \phi\right) = \sum_{i,j} \frac{2}{mn} \frac{\hat{D}_{ij} - \|\tilde{D}_{ij}\|}{\|\tilde{D}_{ij}\|} \tilde{D}_{ij} \odot Y_j \cdot \frac{\partial R_t}{\partial \mathbf{q}_t}, \tag{16}$$

$$\nabla_\mathbf{t} \mathcal{L}_t\left(R_t, \mathbf{t}_t, \phi\right) = \sum_{i,j} \frac{2}{mn} \frac{\hat{D}_{ij} - \|\tilde{D}_{ij}\|}{\|\tilde{D}_{ij}\|} \tilde{D}_{ij},$$

where rotation matrix $R$ and quaternion $\mathbf{q}$ are converted to each other through the Bar-Itzhack algorithm [5], and $\odot$ means Hadamard product. After $T$ rounds of iterations, $(R^*(\phi), \mathbf{t}^*(\phi))$ can be approximated as $(R_T, t_T)$. We can compute the hypergradient by substituting

$$\frac{\partial (R^*(\phi), \mathbf{t}^*(\phi))}{\partial \phi} \approx -\gamma \frac{\partial^2 \mathcal{L}(R_T, \mathbf{t}_T, \phi)}{\partial (R_T, \mathbf{t}_T)^\top \partial \phi}. \tag{17}$$

Due to the vast search space and the rugged landscape of the loss, the gradient descent algorithm with random initialization often requires a large number of iterations to converge and struggles to find the global minima. In the next subsection, we will derive a spectral initialization to enhance convergence.

## 3.3 Spectral Initialization

Recall that we intend to derive a good initialization $(R_0, \mathbf{t}_0)$ for the gradient descent of Equation (1). To simplify it, we denote $\hat{Y} = RY - \mathbf{t}\mathbf{1}_n^\top$, where $\mathbf{1}_n$ is the all-ones vector. Then substitute $\hat{Y}$ into

$$\left(\|X_i - RY_j - \mathbf{t}\| - \hat{D}_{ij}\right)^2 = X_i^\top X_i - 2X_i^\top \hat{Y}_j + \hat{Y}_j^\top \hat{Y}_j + \hat{D}_{ij}^2 - 2\hat{D}_{ij}\|X_i - \hat{Y}_j\|. \tag{18}$$

According to these variable combinations, we define four variables and two centering matrices

$$B = \begin{bmatrix} X_1^\top X_1 & \dots & X_1^\top X_1 \\ X_2^\top X_2 & \dots & X_2^\top X_2 \\ \dots & \dots & \dots \\ X_m^\top X_m & \dots & X_m^\top X_m \end{bmatrix}_{m \times n} \quad C = \begin{bmatrix} \hat{Y}_1^\top \hat{Y}_1 & \dots & \hat{Y}_n^\top \hat{Y}_n \\ \hat{Y}_1^\top \hat{Y}_1 & \dots & \hat{Y}_n^\top \hat{Y}_n \\ \dots & \dots & \dots \\ \hat{Y}_1^\top \hat{Y}_1 & \dots & \hat{Y}_n^\top \hat{Y}_n \end{bmatrix}_{m \times n}$$

$$E = X^\top \hat{Y} = \begin{bmatrix} X_1^\top \hat{Y}_1 & \dots & X_1^\top \hat{Y}_n \\ X_2^\top \hat{Y}_1 & \dots & X_2^\top \hat{Y}_n \\ \dots & \dots & \dots \\ X_m^\top \hat{Y}_1 & \dots & X_m^\top \hat{Y}_n \end{bmatrix}_{m \times n} \quad F = \begin{bmatrix} \hat{D}_{11}^2 & \dots & \hat{D}_{1n}^2 \\ \hat{D}_{21}^2 & \dots & \hat{D}_{2n}^2 \\ \dots & \dots & \dots \\ \hat{D}_{m1}^2 & \dots & \hat{D}_{mn}^2 \end{bmatrix}_{m \times n} \tag{19}$$

$$H_m = I_m - \frac{1}{m} J_m \quad H_n = I_n - \frac{1}{n} J_n,$$

where $I$ represents the identity matrix and $J$ refers to the all-ones matrix. If the condition $\forall i, j, \|X_i - \hat{Y}_j\| = \hat{D}_{ij}$ holds, the following equation exists

$$H_m F H_n = H_m (B - 2E + C) H_n = -2 H_m E H_n = -2 H_m X^\top R Y H_n. \tag{20}$$

Table 1: Statistics of datasets.

| Datasets | # Pairs of Proteins | # Residues per Protein | # Atoms per Protein |
|---|---|---|---|
| **Training Set** | 4890 | 565.9 ($\pm$264.9) | 4334.4 ($\pm$2028.3) |
| **DB5.5 (Test)** | 24 | 428.4 ($\pm$132.0) | 3308.0 ($\pm$1000.5) |
| **VH-VL (Test)** | 68 | 230.1 ($\pm$5.4) | 1749.9 ($\pm$53.7) |
| **AB-AG (Test)** | 68 | 433.0 ($\pm$72.0) | 3346.7 ($\pm$568.8) |

By performing singular value decomposition (SVD) for $H_m X^\top$ and $Y H_n$ respectively

$$H_m X^\top = U_X \Sigma_X V_X^\top, \quad Y H_n = U_Y \Sigma_Y V_Y^\top, \tag{21}$$

the rotation matrix can be solved as

$$R = -\frac{1}{2} V_X \Sigma_X^{-1} U_X^\top H_m F H_n V_Y \Sigma_Y^{-1} U_Y^\top. \tag{22}$$

Given the rotation matrix $R$, the translation vector $\mathbf{t}$ can be gotten using

$$H_m F = H_m (B - 2E + C) = H_m B - 2 H_m X^\top R Y - 2 H_m X^\top \mathbf{t} \mathbf{1}_n^\top, \tag{23}$$

$$\mathbf{t} = -\frac{1}{2} V_X \Sigma_X^{-1} U_X^\top H_m (F - B + 2 X^\top \hat{R} Y) \mathbf{1}_n. \tag{24}$$

By replacing random initialization with the above spectral initialization, the gradient descent will approach the global minima faster and better. We will empirically verify this conclusion through subsequent experiments.

## 4 Experiments

**Datasets** We leverage Docking Benchmark 5.5 (DB5.5) [52], a gold standard dataset tailored for rigid docking. Following the experimental setting of EquiDock [25], the dataset is randomly partitioned into a test split of size 24. For a comprehensive comparison, we curate two datasets of antibodies (VH-VL) and antibody-antigen complexes (AB-AG) from Protein Data Bank (PDB) [3] and expect them to become new benchmarks. Specifically, the training set consists of 4,890 complexes containing at least one antibody chain, while the test set comprises 68 antibody-antigen complexes released after October 2022. The docking results of variable heavy-light chains and antigen-antibody can be separately evaluated. We direct the readers of interest to Appendix A for extraction criteria and detailed identifier lists. The statistics of the dataset are summarized in Table 1.

**Evaluation Protocol** We compare BiDock with two categories of representative methods, including three docking software ZDOCK [13], ClusPro [29], and HDOCK [58], and two deep learning models EquiDock [25], and AlphaFold-Multimer (Multimer for short [23]). To measure the quality of

Table 2: Quantitative results on protein docking. (bold: best; underline: runner-up)

| Datasets | Metrics | ZDOCK | ClusPro | HDOCK | EquiDock | Multimer | BiDock |
|---|---|---|---|---|---|---|---|
| **DB5.5** | *RMSD* $\downarrow$ | 12.491$_{\pm6.294}$ | 14.135$_{\pm8.153}$ | 11.328$_{\pm8.073}$ | 14.982$_{\pm5.304}$ | 7.797$_{\pm7.428}$ | **7.280**$_{\pm8.117}$ |
| | *TM-score* $\uparrow$ | 0.689$_{\pm0.114}$ | 0.702$_{\pm0.118}$ | 0.742$_{\pm0.167}$ | 0.714$_{\pm0.114}$ | 0.821$_{\pm0.162}$ | **0.847**$_{\pm0.158}$ |
| | *DockQ* $\uparrow$ | 0.084$_{\pm0.113}$ | 0.118$_{\pm0.192}$ | 0.314$_{\pm0.390}$ | 0.030$_{\pm0.029}$ | 0.469$_{\pm0.396}$ | **0.564**$_{\pm0.369}$ |
| **VH-VL** | *RMSD* $\downarrow$ | 10.982$_{\pm3.864}$ | 5.899$_{\pm5.688}$ | 2.032$_{\pm2.388}$ | 18.293$_{\pm2.871}$ | 1.325$_{\pm0.530}$ | **1.242**$_{\pm0.602}$ |
| | *TM-score* $\uparrow$ | 0.596$_{\pm0.075}$ | 0.792$_{\pm0.156}$ | 0.926$_{\pm0.100}$ | 0.559$_{\pm0.017}$ | 0.962$_{\pm0.020}$ | **0.966**$_{\pm0.021}$ |
| | *DockQ* $\uparrow$ | 0.108$_{\pm0.134}$ | 0.404$_{\pm0.277}$ | 0.705$_{\pm0.201}$ | 0.032$_{\pm0.016}$ | 0.765$_{\pm0.094}$ | **0.773**$_{\pm0.187}$ |
| **AB-AG** | *RMSD* $\downarrow$ | 18.892$_{\pm3.757}$ | 15.670$_{\pm6.896}$ | 15.779$_{\pm6.364}$ | 18.468$_{\pm2.706}$ | 13.650$_{\pm5.886}$ | **9.707**$_{\pm8.759}$ |
| | *TM-score* $\uparrow$ | 0.504$_{\pm0.059}$ | 0.596$_{\pm0.143}$ | 0.612$_{\pm0.137}$ | 0.502$_{\pm0.096}$ | 0.640$_{\pm0.124}$ | **0.773**$_{\pm0.187}$ |
| | *DockQ* $\uparrow$ | 0.035$_{\pm0.031}$ | 0.108$_{\pm0.181}$ | 0.090$_{\pm0.187}$ | 0.043$_{\pm0.017}$ | 0.108$_{\pm0.172}$ | **0.342**$_{\pm0.351}$ |
| | *maxDockQ* $\uparrow$ | 0.042$_{\pm0.043}$ | 0.136$_{\pm0.220}$ | 0.111$_{\pm0.237}$ | 0.043$_{\pm0.018}$ | 0.125$_{\pm0.214}$ | **0.414**$_{\pm0.386}$ |

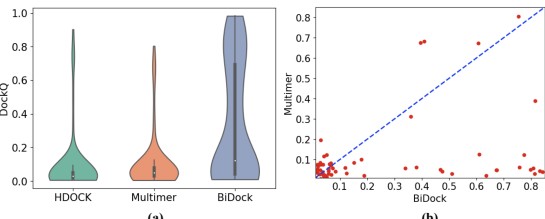

| | (a) | | (b) |

Figure 4: An intuitive comparison of DockQ metric on antibody-antigen docking. The box inside (a) violin plot represents 25-75 percentiles, and the median is shown by a white dot. Scatters in (b) scatter plot appear under the dashed diagonal line, indicating that BiDock outperforms Multimer on these complexes.

Table 3: Total inference time of different methods on antibody-antigen docking. (unit: hour)

| Methods | Inference time |
|---------|----------------|
| ZDOCK | 72.61 |
| ClusPro | 87.27 |
| HDOCK | 20.40 |
| EquiDock | 0.60 |
| Multimer | 1.07 |
| BiDock | 1.47 |

predictions, we report universally accepted metrics Root Mean Square Deviation (RMSD), TM-score (Template Modeling score), and DockQ [6]. Please note that in the context of antibody-antigen docking, the original DockQ metric evaluates the docking performance by treating the entire antibody as a single entity to the antigen. Furthermore, we can assess the docking results separately for variable heavy/light chain (VH/VL) to the antigen and denote the maximum value as maxDockQ. Refer to Appendix B for details of experiments, including implementation and hyperparameters.

## 4.1 Main Results and Analysis

**Docking Results** Table 2 demonstrates that BiDock generally produces acceptable predictions on all datasets. Notably, BiDock achieves a significant improvement in performance on the more challenging antibody-antigen docking, with a 234% relative gain over the runner-up on the DockQ metric. These results highlight the effectiveness of our bi-level optimization, which effectively leverages multimodal information. We also observe that some established docking software, such as HDOCK, still provides reliable predictions. In contrast, deep learning methods may have some leeway in performance. For instance, the mean and deviation of RMSD evaluated from EquiDock are relatively large, indicating that some inappropriate SE(3) transformations are learned.

To be more intuitive, we select representative baselines, HDOCK and Multimer, along with BiDock to analyze their performance differences on antibody-antigen docking. The distribution of DockQ for each method is displayed using a violin plot, and a direct comparison between Multimer and BiDock is also drawn in Figure 4. It reveals that the distribution of DockQ for BiDock is more concentrated around high values, indicating a higher percentage of successful docking predictions. Overall, these results suggest that our proposed BiDock is a promising approach for rigid protein docking.

**Visualization** To visually demonstrate the superiority of our proposed BiDock, we chose the spike glycoprotein $7F6Z$ as an example antigen. The spike glycoprotein is crucial for the entry of coronaviruses into host cells, making it a target of great interest for therapeutic intervention and vaccine development. In Figure 5, we align the ground truth structure of this protein with predictions from competitive baselines. Upon inspection, it is evident that predictions from baselines exhibit noticeable deviations from the ground truth structures. Such inaccuracies can significantly impact our understanding of the binding mechanisms and hinder the design of effective interventions. In contrast,

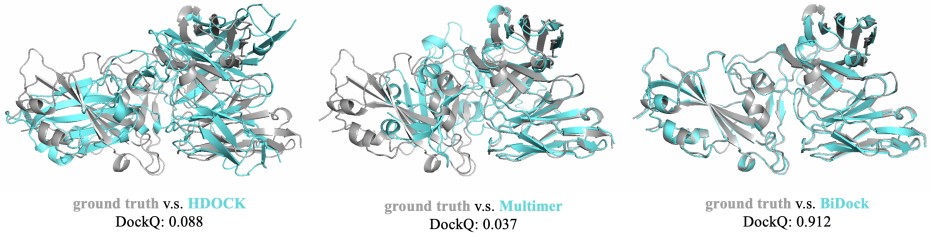

Figure 5: Structure comparison between predictions and the ground truth of protein complex $7F6Z$. The ground truth structures are represented in light gray, while predictions are colored cyan.

Table 4: Ablation studies on spectral initialization, bi-level optimization, and masked MSA loss. (number in parentheses: gradient descent steps; unit: thousand; bold: best)

|  | RMSD | DockQ |
|---|---|---|
| **w/o SI(1)** | $14.140_{\pm 8.032}$ | $0.187_{\pm 0.256}$ |
| **w/o SI(2)** | $11.191_{\pm 8.754}$ | $0.290_{\pm 0.328}$ |
| **w/o SI(4)** | $9.806_{\pm 8.646}$ | $0.335_{\pm 0.347}$ |
| **w/o Bi(2)** | $10.090_{\pm 7.817}$ | $0.220_{\pm 0.232}$ |
| **w/o MM(2)** | $9.821_{\pm 8.688}$ | $0.330_{\pm 0.350}$ |
| **BiDock(2)** | $\mathbf{9.707}_{\pm 8.759}$ | $\mathbf{0.342}_{\pm 0.351}$ |

Table 5: Impacts of different optimizers on antibody-antigen docking. (number in parentheses: gradient descent steps; unit: thousand; bold: best)

|  | RMSD | DockQ |
|---|---|---|
| **SGD(1)** | $9.939_{\pm 8.525}$ | $0.333_{\pm 0.341}$ |
| **SGD(2)** | $9.940_{\pm 8.557}$ | $0.335_{\pm 0.344}$ |
| **SGD(5)** | $9.914_{\pm 8.556}$ | $0.337_{\pm 0.345}$ |
| **SGD(10)** | $9.861_{\pm 8.548}$ | $0.337_{\pm 0.344}$ |
| **Adam(1)** | $9.780_{\pm 8.773}$ | $0.341_{\pm 0.350}$ |
| **Adam(2)** | $\mathbf{9.707}_{\pm 8.759}$ | $\mathbf{0.342}_{\pm 0.351}$ |

BiDock successfully captures the correct docking interface and accurately predicts the binding pose, highlighting its potential for providing biological insights and aiding drug design.

**Computational Efficiency**   Table 3 shows the total inference time for antibody-antigen complexes. Traditional docking software, involving candidate sampling, ranking, and refinement steps, incurs substantial computational costs. Fortunately, deep learning methods provide a significant speed-up, which is particularly important in efficient screening. Although EquiDock is the fastest, its performance falls short of traditional software due to limitations in leveraging coevolution information and simple networks. On the other hand, Multimer and BiDock exhibit comparable inference times. Considering the performance improvement of our model, this trade-off is acceptable.

## 4.2   Ablation Studies

**Effects of Spectral Initialization**   Recalling our utilization of spectral estimation to derive a numerical solution for initializing the gradient descent in the inner loop, we conduct ablation experiments to demonstrate the effectiveness of this spectral initialization. Table 4 presents the results on antibody-antigen docking, where the variant without spectral initialization is denoted as "w/o SI" and the number of gradient descent steps is listed in parentheses. It is seen that the "w/o SI" variants exhibit slower convergence and lower accuracy, even when we further increase the number of gradient descent steps. These findings further support the benefits of our proposed spectral initialization in accelerating the optimization process and achieving superior results.

**Effects of Bi-level Optimization**   To justify the effectiveness of bi-level optimization, we use a two-stage strategy: training outer and inner loops separately. Specifically, we use cross-entropy and masked MSA losses to train cross-modal transformer. Based on the resulting distance map, we compute roto-translation transformation with spectral initialization. The results on antibody-antigen docking are presented in Table 4, where "w/o Bi" denotes the variant without bi-level optimization (results reported from the original paper [34]). Results support our contributions of employing bi-level optimization for end-to-end optimization, which customizes the parameter learning of the cross-modal transformer for rigid docking.

**Effects of Masked MSA Loss**   In addition to the essential loss for the distance map, we introduce a masked MSA loss in the outer loop to supervise the learning of evolution representations. We also conduct an ablation study to investigate its significance, as shown in Table 4. The "w/o MM" refers to models without masked MSA loss. According to the results, it can be concluded that the masked MSA loss enables the cross-modal transformer to more effectively leverage the rich evolutionary information and seamlessly integrate it with the structure-modal information.

**Effects of Optimizer**   As shown in Equation (15), we have the flexibility to choose different optimizers for the inner loop. To explore the impact of different optimizers on convergence speed and performance, we compare the effects of SGD and Adam optimizers. We vary the number of

gradient descent steps and evaluate the final performance, as displayed in Table 5. With Adam, BiDock achieves faster convergence and attains higher accuracy, indicating that a better optimizer can navigate the landscape and find better minima during the optimization of the inner loop.

## 5    Conclusion

In this study, we introduce BiDock, a novel model for rigid protein docking that tackles the challenge of accurately predicting the 3D structure of protein complexes from unbound states. By formulating this problem as a bi-level optimization, BiDock combines the advantages of integrating multimodal information by a cross-modal transformer in the outer loop and maintaining the rigidity of learning roto-translation transformation in the inner loop. Additionally, we derive a spectral initialization to expedite convergence. The maximum 234% relative performance improvement validates the effectiveness of BiDock in rigid protein docking.

**Limitations and Broader Impact**    Despite the encouraging results, BiDock does not explicitly incorporate geometric constraints between residues when learning the distance map and does not account for potential atom clashes. Our future work will address these limitations and extend our framework to general proteins. Protein docking deepens our understanding of biological mechanisms and aids in the design of targeted interventions. This research may inspire the AI4Science community to pay more attention to the practical challenges in docking and promote further advancements in this field with significant real-world implications.

## Acknowledgments and Disclosure of Funding

**Acknowledgments**   We would like to extend our heartfelt appreciation to the anonymous reviewers who dedicated their valuable time and expertise to meticulously review this paper. Their constructive feedback significantly contributed to the refinement and enhancement of the quality of our work. Our gratitude also extends to our team members, whose insightful discussions and collaborative brainstorming sessions were instrumental in shaping this research.

We wish to express our sincere thanks to the Data Science team at BioMap, especially to Tingting Tang, Nachuan Shan, Shiro Hong, Di Wang, Dacheng Li, Chenrui Xu, Ke Wang, and Cheng Chen, for their exceptional support in data collection, processing, and analysis. Their efforts with complex biological datasets were indispensable to the success of our research. Furthermore, we also express our sincere thanks to the AI Infracture team at BioMap, especially to Ming Yang, Chuang Yu, Zedong Zheng, Zezhi Wang, Qian Wang, and Xiaoming Zhang, for their outstanding technical support and seamless collaboration. Their proactive approach and effective troubleshooting played a vital role in ensuring the smooth progress of our research.

Finally, thanks Aaron for proofreading the manuscript.

**Author Contributions**   The author's contributions to this paper encompass a range of vital aspects. Ruijia Wang made substantial contributions to content, writing, and experiment design. Yiwu Sun took on a central role in implementing the bi-level optimization algorithm and conducting experiments. Yujie Luo and Shaochuan Li were crucial in developing the precursor algorithm, xTrimoDock [34], and also aiding in data preparation.

Le Song and Hui Li lead the antibody-antigen structure prediction project in BioMap. They made invaluable contributions by providing algorithmic insights and guidance to the project, influencing the overall conceptualization and direction of the research. Xingyi Cheng, Chuan Shi and Cheng Yang provided valuable guidance in organizing the writing of the paper.

This work was funded by BioMap. The intellectual property of BiDock is 100% owned by BioMap.

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

# A  Details of Dataset

**Background of Antibodies**    Antibodies are
vital components of the immune system and
are classified into various classes, including
IgG, IgM, IgA, IgD, and IgE. Among them,
IgG antibodies are the most abundant in the
bloodstream and play a primary role in im-
mune responses against pathogens.

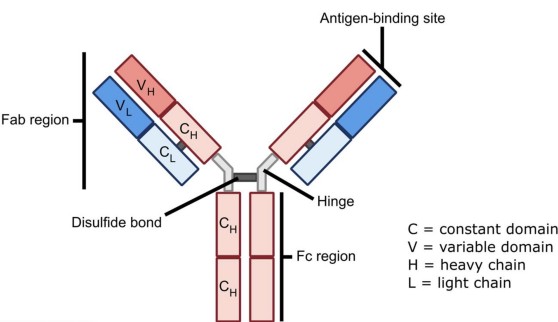

As depicted in Figure 6, IgG antibodies ex-
hibit a Y-shaped structure composed of two
identical light chains and two identical heavy
chains, where heavy chains provide struc-
tural stability. Each antibody chain is further
divided into distinct regions. (1) The vari-
able regions, referred to as the variable heavy
(VH) and variable light (VL) regions, are lo-

Figure 6: Structure of an IgG antibody. The heavy
chain is colored orange, while the light chain is blue.

cated at the tips of the Y arms. These regions contribute to the specificity of antibodies in recognizing
and binding to antigens. The VH and VL regions collaborate to form the fragment antigen-binding
(Fab) region. (2) At the base of the Y structure, the constant regions, also known as the fragment
crystallizable (Fc) region, are important in the effector functions of antibodies. The Fc region interacts
with immune cells and triggers immune responses, such as the activation of complement proteins for
pathogen destruction and the promotion of phagocytosis.

Given this background knowledge of antibodies, it becomes clear that antibody-antigen docking is
fundamental in immune responses, therapeutic applications, vaccine development, and drug discovery.
Therefore, our study places a particular emphasis on antibody-antigen docking, contributing to
this field by curating a high-quality benchmark. This dataset will serve as a valuable resource for
evaluating computational models in predicting antibody-antigen interactions, ultimately facilitating
the development of novel therapeutics and immunological interventions.

**Antibody-antigen Benchmark**    The training set comprises 4,890 complexes of antibody-antigen
pairs, each consisting of proteins with a minimum of 30 residues. These complexes encompass
three chains, including the light and heavy chains of the antibody, along with one antigen chain. All
complexes were released before January 2022. Similarly, the test set consists of 68 antibody-antigen
complexes with three chains, released after October 2022. Thus, we ensure that neither baselines nor
our proposed model was trained using the test set and avoid data leakage.

In practical applications, obtaining the ground truth structures of antibody-antigen complexes poses
significant challenges. Researchers often turn to existing folding models to predict them. To simulate
real-world scenarios, we employ a specialized antibody model called xTrimoABFold [55, 56, 54]
to predict the conformations of antibodies and AlphaFold2 [28] for antigens. Given these predicted
structures as rigid structures, we construct training and test datasets essential for further analysis and
investigation. Here are the PDB identifiers and their corresponding chain identifiers for the test set.

8dls:A,H,L; 8dlr:A,H,L; 8dfi:A,H,L; 8dfh:A,H,L; 8dcc:E,L,H; 8dad:H,L,B; 7zr8:A,H,L; 7zf8:H,L,E;

7xxl:C,A,B; 7xh8:A,B,C; 7x26:H,I,K; 7wsl:L,D,H; 7wsi:A,H,L; 7ws6:C,H,I; 7ws2:D,A,E;

7wrz:H,L,R; 7wrv:C,U,V; 7wro:H,L,R; 7wrl:A,B,R; 7wrj:R,A,B; 7wog:C,A,B; 7wlc:E,H,L;

7wef:C,E,L; 7wee:E,H,L; 7wed:E,H,L; 7wcr:A,a,b; 7wbz:A,H,L; 7urq:A,H,L; 7uaq:A,H,L;

7tty:A,H,L; 7ttx:A,H,L; 7ttm:A,H,L; 7tpj:B,H,L; 7tp4:Z,H,L; 7tp3:Z,H,L; 7tlz:A,B,J;

7the:A,B,C; 7tc9:B,H,L; 7t8w:L,H,D; 7t7b:A,H,L; 7t01:A,H,L; 7swp:A,H,L; 7su1:H,L,C;

7str:L,H,C; 7sem:B,F,C; 7sd5:A,H,L; 7sbu:A,H,L; 7sbg:H,L,C; 7sbd:H,L,C; 7sa6:A,H,L;

7s5p:A,H,L; 7rxp:L,H,A; 7rxi:L,H,A; 7rbu:B,H,L; 7qtk:D,B,C; 7n0a:C,A,B; 7lo8:Z,H,L;

7lo7:Z,H,L; 7kql:H,L,T; 7fjc:H,L,E; 7f7e:C,L,E; 7f6z:R,H,L; 7f6y:R,H,L; 7eng:L,H,B;

7ek0:R,H,L; 7ejz:R,H,L; 7ejy:R,H,L; 7e9p:B,L,H

# B   Details of Implementation

**Baselines**   ZDOCK[5], ClusPro[6], and HDOCK[7] are user-friendly local packages suitable for automated experiments or web servers for manual submissions. We select the top-1 predicted structure from each of these methods for subsequent evaluation. For EquiDock[8] and Multimer[9], we utilize their pretrained models available on GitHub for the inference. It is worth emphasizing that all methods except Multimer are designed for docking two chains. Therefore, during the evaluation, we employ a sequential docking strategy. This entails initially docking the light chain and heavy chain together, followed by treating them as a unified entity for docking with the antigen. And we calculate evaluation metrics using the tools USalign[10] and DockQ[11].

**MSA Extraction**   We utilize the heuristic approach described in [23] to pair sequences from per-chain multiple sequence alignments (MSAs). Initially, the per-chain MSA sequences are grouped based on species, with the species labels obtained from UniProt's idmapping [12]. Within each specific species group, the sequences are paired together. We match the chain MSAs by minimizing the base-pair distance between the chains for prokaryotic species. While in terms of eukaryotic species, we order them based on sequence identity to the target sequence [62]. To reduce computational and memory costs, we employ the MSA clustering approach from AlphaFold2 [28]. We randomly select $N_{cls} = 252$ sequences as the MSA cluster centers, with the primary protein sequence always set as the first cluster center. The remaining sequences are assigned to their closest cluster based on the Hamming distance.

**Sequence-modal Input**   The sequence modality incorporates information derived from the primary sequence itself and co-evolutionary information obtained from MSAs. Following prior research [28, 23], we extract two types of features: type features $F^{typ} \in \mathbb{R}^{N_{res} \times 21}$ and primary pair features $F^{pp} \in \mathbb{R}^{N_{res} \times N_{res} \times 73}$ from the primary sequence, where $N_{res}$ represents the number of residues. Regarding MSAs, we utilize cluster MSA features $F^{msa} \in \mathbb{R}^{N_{cls} \times N_{res} \times 49}$, where $N_{cls}$ denotes the number of cluster centers. Specifically,

- The *type feature* $F^{typ} \in \mathbb{R}^{N_{res} \times 21}$ comprises one-hot representations of the amino acid types, encompassing the 20 known amino acids and one additional category for unknown types.

- The *primary pair feature* $F^{pp} \in \mathbb{R}^{N_{res} \times N_{res} \times 73}$ contains positional information within or across chains, including three components. (1) The *relative positional feature* of size $[N_{res}, N_{res}, 66]$ represents the relative residue indices, which are clipped between $[-32, 32]$. The 66-th index is used to indicate cross-chain pairs. (2) The *entity indicator* of size $[N_{res}, N_{res}, 1]$ identifies whether residues $i$ and $j$ originate from the same chain. (3) The *relative index feature* of size $[N_{res}, N_{res}, 6]$ introduces the relative *sym_id*[13] indices clipped between $[-2, 2]$. The 6-th index is assigned to pairs where the two residues have different *sym_id*s.

- The *cluster MSA feature* $F^{msa} \in \mathbb{R}^{N_{cls} \times N_{res} \times 49}$ consists of five components. (1) The *one-hot representation of the amino acid types* with size $[N_{cls}, N_{res}, 23]$, including 20 amino acids, one unknown type, one gap or missing residue, and one mask token as introduced in Section 3.1. (2) The *amino acid distribution* of size $[N_{cls}, N_{res}, 23]$ represents the distribution of amino acid types within each MSA cluster. (3) The *deletion indicator* of size $[N_{cls}, N_{res}, 1]$ indicates whether there is a deletion to the left of each residue. (4) The *deletion value* of size $[N_{cls}, N_{res}, 1]$ is calculated using the formula $\frac{2}{\pi} \arctan \frac{c}{3}$, where $c$ refers to the number of deletions to the left of each position.

---

[5]`https://zdock.umassmed.edu`

[6]`https://cluspro.org`

[7]`http://hdock.phys.hust.edu.cn`

[8](MIT license) `https://github.com/octavian-ganea/equidock_public`

[9](Apache-2.0 license) `https://github.com/aqlaboratory/openfold`

[10](MIT license) `https://github.com/pylelab/USalign`

[11](GPL-3.0 license) `https://github.com/bjornwallner/DockQ`

[12]`https://ftp.uniprot.org/pub/databases/uniprot/current_release/knowledgebase/idmapping`

[13]The *sym_id* is used to distinguish chains with the same sequence. For example, we consider a complex comprising five chains $\{A, B, B, C, C\}$, where $A$, $B$, and $C$ represent three unique chains. The corresponding *sym_id*s for each chain would be $\{1, 1, 2, 1, 2\}$, respectively.

(5) The *mean deletion value* of size $[N_{cls}, N_{res}, 1]$ is computed as $\frac{2}{\pi} \arctan \frac{\bar{c}}{3}$, where $\bar{c}$ represents the average number of deletions to all residues on the left of each position.

**Structure-modal Input**  For the structure modality, we extract angle features $F^{ang} \in \mathbb{R}^{N_{res} \times 57}$ and pair features $F^p \in \mathbb{R}^{N_{res} \times N_{res} \times 88}$ from the rigid protein structures. These features capture important structure-modal information and are used as input for our docking model. Specifically,

- The *angle feature* $F^{ang} \in \mathbb{R}^{N_{res} \times 57}$ comprises three components. (1) The *one-hot representation of the amino acid types* with a size of $[N_{res}, 22]$, including 20 amino acids, one unknown type, and one gap or missing residue. (2) The *angle representations* of size $[N_{res}, 28]$ use sine and cosine to encode three backbone torsion angles, four side-chain torsion angles, and alternative torsion angles with $180°$ rotation symmetry for each local frame of residue. (3) The *angle indicator* with size $[N_{res}, 7]$ indicates the presence or absence of torsion angles.

- The *pair feature* $F^p \in \mathbb{R}^{N_{res} \times N_{res} \times 88}$ comprises five components. (1) The *distogram feature* of size $[N_{res}, N_{res}, 39]$ represents the discretized distances between $C\beta$ atoms. In the case of glycine, which lacks $C\beta$ atoms, $C\alpha$ is used instead. The distances are discretized into 38 bins of equal width ranging from 3.25 to 50.75Å, with an additional bin accounting for larger distances. (2) The *residue type feature* of size $[N_{res}, N_{res}, 44]$ is derived from expanding one-hot representations of residue types with dimensions of $[N_{res}, 1, 22]$ and $[N_{res}, 22, 1]$. (3) The *backbone feature* of size $[N_{res}, N_{res}, 3]$ is obtained by constructing the unit vector of the local frame through the Gram-Schmidt process based on the original N-C$\alpha$-C coordinates. (4) The *residue indicator* with size $[N_{res}, N_{res}, 1]$ is expanded from the indicator of residue existence. (5) The *pair indicator* of size $[N_{res}, N_{res}, 1]$ indicates whether the pair is masked.

**MSA Mask Policy**  Reflecting on Section 3.1, we design a masked MSA loss to supervise the learning of evolution representations and the integration of cross-modal information. Specifically, we randomly mask each position in an MSA cluster center with a 15% probability. Each masked token is replaced according to the following policies:

- 70% probability of substitution with a special token $\star$
- 10% probability of substitution with a randomly selected amino acid from a uniform distribution
- 10% probability of substitution with an amino acid sampled from the MSA profile that corresponds to the position
- 10% probability of no substitution

**Hyperparameter Settings**  We initialize specific parameters of the cross-modal transformer with the checkpoint of Multimer and implement bi-level optimization using TorchOpt[14] library. The crop size is set to 412, and the batch size is set to 1. The coefficients in Equation (13) are $\lambda_1 = 0.2$, $\lambda_2 = 2.0$, and $\lambda_3 = 10.0$. For optimization, we employ the Adam optimizer with a learning rate of $10^{-4}$ and integrate learning rate warmup, gradually increasing the learning rate from 0 to $10^{-4}$ within the first 100 steps. The exponential moving average (EMA) strategy applies a decay rate of $\beta = 0.999$ and undergoes updates every 200 steps. The environment where we run all experiments is:

- Operating system: Linux version 5.13.0-30-generic
- CPU information: AMD EPYC 7742 64-Core Processor
- GPU information: NVIDIA A100-SXM4-80GB

## C  Additional Results

**Effects of Noisy Structures**  Classical software rely on score functions derived from statistics in the protein data bank. This dependency renders them susceptible to noise. When using folding algorithms to predict unbounded proteins, the performance of these software can degrade significantly. To validate this intuition, we conduct a docking performance analysis on the DB5.5 dataset using ground truth and predicted structures from folding models as unbounded structures, respectively. As shown in Table 6, these results illustrate that although HDOCK performs exceptionally well with

---

[14](Apache-2.0 license) `https://github.com/metaopt/torchopt`

Table 6: Impacts of noisy structures on the docking performance of classical software and BiDock. (bold: best; underline: runner-up)

| | Ground Truth | | | Predicted Structure | | |
|---|---|---|---|---|---|---|
| | *RMSD* ↓ | *TM-score* ↑ | *DockQ* ↑ | *RMSD* ↓ | *TM-score* ↑ | *DockQ* ↑ |
| **ZDOCK** | $11.830_{\pm 5.227}$ | $0.738_{\pm 0.120}$ | $0.095_{\pm 0.130}$ | $12.491_{\pm 6.294}$ | $0.689_{\pm 0.114}$ | $0.084_{\pm 0.113}$ |
| **ClusPro** | $11.486_{\pm 7.993}$ | $0.780_{\pm 0.133}$ | $0.204_{\pm 0.256}$ | $14.135_{\pm 8.153}$ | $0.702_{\pm 0.118}$ | $0.118_{\pm 0.192}$ |
| **HDOCK** | $\mathbf{3.464}_{\pm 7.394}$ | $\mathbf{0.935}_{\pm 0.144}$ | $\mathbf{0.815}_{\pm 0.364}$ | $\underline{11.328}_{\pm 8.073}$ | $\underline{0.742}_{\pm 0.167}$ | $\underline{0.314}_{\pm 0.390}$ |
| **BiDock** | $\underline{6.173}_{\pm 8.825}$ | $\underline{0.892}_{\pm 0.156}$ | $\underline{0.648}_{\pm 0.432}$ | $\mathbf{7.280}_{\pm 8.117}$ | $\mathbf{0.847}_{\pm 0.158}$ | $\mathbf{0.564}_{\pm 0.369}$ |

ground truth, minor noise in predicted structures leads to a substantial decline in its performance. On the contrary, BiDock consistently generates acceptable predictions regardless of the input type, showcasing its robustness to noise. In real-world applications, reliance on the availability of ground truth structures is impractical. The ability of BiDock to maintain high prediction quality when confronted with noisy structures makes it an invaluable tool.

