# OpenReview forum: "Injecting Multimodal Information into Rigid Protein Docking via Bi-level Optimization"
_NeurIPS.cc/2023/Conference — NeurIPS 2023 poster_

### Official Review · Reviewer_HJaR · 2023-07-04

**Soundness:** 3 good
**Presentation:** 3 good
**Contribution:** 3 good
**Rating:** 7
**Confidence:** 4

**Summary:**

The paper proposes a new framework for rigid protein docking called BiDock. BiDock is based on an Evoformer-based model that predicts a distance matrix and a gradient-based optimization of the optimal roto-translation based on the predicted distance matrix. In order to be able to pass gradients through the optimization steps at training time the authors derive a bi-level optimization procedure and a spectral initialization.

The model is trained specifically for rigid docking of antibody chains and evaluated on a range of complexes divided in: general proteins, antibody heavy and light chains and antibody-antigens. Across all three categories, the method shows improved performances.

**Strengths:**

The paper proposes an interesting new approach to deep-learning-based rigid docking using gradient based pose optimization and propagation through the optimization. Moreover, the spectral initialization provides an interesting and meaningful improvement to the performance.

**Weaknesses:**

The paper has good methodological and empirical contributions, however, to make it a strong accept I believe a few components in the manuscript should be improved and/or clarified.

Introduction and novelty claims:

1. Personally, I do not understand the strong emphasis that the authors put on “being the first to effectively leverage sequence and structure modal information in rigid protein docking”. This is I believe the least significant part of the paper as: (1) AlphaFold-Multimer already uses sequence and structural template based informations and the architecture proposed is very similar to that of the EvoFormer blocks, (2) adding sequence based information to EquiDock would be a very straightforward addition, (3) other existing deep-learning based docking methods not referenced by the authors such as [1] already uses sequence and structure informations for the predictions. I believe that the claims of novelty in these regards should be adjusted taking into consideration (1-3).

Unclear parts of the method:

2. The loss used to train the model is not clear to me. Equation (12) says that L^out(R, t, \phi)  = \lambda_1 L_dist + \lambda_2 L_msa, however neither L_dist nor L_msa seem to depend on R nor t. Is this correct? If so I don’t understand the point of the bi-level optimization, as the gradient of L^out with respect to \phi would not depend on R* or t*.
3. line 167 “\hat{D} can be obtained by using the center of each bin”, what is meant specifically by center of each bin? Center of the bin with highest likelihood, or median of the predicted binned distribution, or something else?

There are several missing details (both in paper and in appendix) about the evaluation set-up that should be clarified:

4. How was AlphaFold-Multimer run (and with what set-up in terms of MSA etc)? The appendix links to the OpenFold repo but this does not yet provide Multimer’s replicas.
5. For the other methods, what structures were used as input? I know DB5.5 provides also apo (unbound) structures that I assume the authors used for the input, however for the two benchmarks they propose how are the input structures generated?
6. I assume the different methods were run on different hardware (GPUs vs CPUs), this should be reported in Table 3 or standardized.

[1] Ketata, Mohamed Amine, et al. "DiffDock-PP: Rigid Protein-Protein Docking with Diffusion Models." *arXiv preprint* (2023).

**Questions:**

See weaknesses section and the following:

7. Especially with the flexible regions in antibodies it appears important to consider a more flexible form of binding, could the proposed framework be extended in that direction?

**Limitations:**

See weaknesses section

---

> ### Author Rebuttal · Authors · 2023-08-09
>
> - We sincerely thank the Reviewer for all the comments, and it is a great honor for us to inspire your interest. We have addressed all your questions below and hope they have clarified all confusion you had about our work.
>
> 1. > I do not understand the strong emphasis that the authors put on “being the first to effectively leverage sequence and structure modal information in rigid protein docking”. There are some related works, such as AlphaFold-Multimer, EquiDock, and DiffDock-PP.
>
> **Answer:** Thanks for your insightful question. (1) AlphaFold-Multimer targets flexible docking, which makes it different from our scope. The main reason is that AlphaFold-Multimer lacks a specific design that allows the direct input of 3D rigid bodies to the structure module for rigid docking. (2) The equivariant GNNs in EquiDock require the feature of each amino acid to be a vector. However, MSA features are tensors with dimensions $F^{msa}\in\mathbb{R}^{N_{cls}\times N_{res}\times 49}$. Thus there is no straightforward extension. (3) Regarding DiffDock-PP, due to space constraints, please refer to the rebuttal provided for Reviewer XfrH's Q1. We apologize for any inconvenience this may have caused you.
>
> In the revised version, we will include these methods in the "Related Work" section, highlighting their differences and reporting additional results in the experiments.
>
> 2. > The loss used to train the model is not clear to me. Equation (12) says that L^out(R, t, \phi) = \lambda_1 L_dist + \lambda_2 L_msa, however neither L_dist nor L_msa seem to depend on R nor t.
>
> **Answer:** Apologies for the confusion. The outer and inner loops share the common goal of learning the optimal roto-translation transformation for rigid docking. Therefore, the inner loss $L^{in}$ in Eq. (1) is also the objective function of the outer loop. To clarify this, we will explicitly include the inner loss $L^{in}$ in Eq. (12) as:
>
> $L^{out}=\lambda_1 L_{dist}+\lambda_2 L_{msa}+\lambda_3 L^{in}$
>
> Thanks for your careful review, and we will fix this error in the revised version.
>
> 3. > Line 167 “\hat{D} can be obtained by using the center of each bin”, what is meant specifically by center of each bin?
>
> **Answer:**  Apologies for the confusion. $\hat{D}$ is obtained by using the mean of each bin. The term "the center of each bin" was not mathematically precise, and we will clarify this point in the revised version. Thanks for your careful review and valuable feedback.
>
> 4. > How was AlphaFold-Multimer run (and with what set-up in terms of MSA etc)?
>
> **Answer:** Apologies for any confusion. We reproduce AlphaFold-Multimer based on AlphaFold2 released by OpenFold and evaluate the reproduced version by converting the official JAX checkpoint (https://github.com/deepmind/alphafold) into PyTorch. The setup of MSA features is the same as those in BiDock, as detailed in Appendix B. Additionally, we feed the rigid bodies of each chain as templates to AlphaFold-Multimer.
>
> Thanks for your careful review. We will include additional information on the experimental setup of AlphaFold-Multimer in the revised version. Your feedback is much appreciated.
>
> 5. > For the other methods, what structures were used as input?
>
> **Answer:** Thanks for your comments. (1) It's worth noting that baselines and our proposed BiDock use the same input. (2) For DB5.5, we utilize the unbound structures (ground truth) provided by the benchmark. Detailed information on the antibody-antigen benchmark is explained in Appendix A. Specifically, obtaining the ground truth structures of antibody-antigen complexes poses significant challenges in practical applications. Researchers often rely on existing folding models to predict them. To simulate real-world scenarios, we use a specialized antibody model called xTrimoABFold [2] to predict the conformations of antibodies and AlphaFold2 for antigens.
>
> Furthermore, our dataset has been selected as a benchmark for the antibody-antigen complex structure prediction task on the "Life Science Leaderboard" (https://www.biomap.com/sota/), facilitating public access and fair comparisons.
>
> 6. > I assume the different methods were run on different hardware (GPUs vs CPUs), this should be reported in Table 3 or standardized.
>
> **Answer:** All experiments for the baselines and our proposed BiDock were conducted on an A100 cluster. The specific environment details are provided in Appendix B as follows:
> - Operating system: Linux version 5.13.0-30-generic
> - CPU information: AMD EPYC 7742 64-Core Processor
> - GPU information: NVIDIA A100-SXM4-80GB
>
> Thanks for your suggestion, and we will clarify this point in the revised version to avoid any misunderstandings.
>
> 7. > Especially with the flexible regions in antibodies it appears important to consider a more flexible form of binding, could the proposed framework be extended in that direction?
>
> **Answer:** Thanks for your insightful question. Rigid docking has practical applications that motivated our focus on this problem. Specifically, (1) rigid docking allows us to input partially known structures, facilitating the solution of challenging docking problems. (2) In certain scenarios, we only need to determine whether a protein exhibits a specific function, focusing on the docking pocket without requiring detailed interface information. In such cases, flexible docking is redundant. (3) Regarding the extension to flexible docking, a well-optimized distance map (cross-modal transformer) and the overall bi-level optimization serve as a good starting point. We can design an inner loop tailored for flexible docking, considering conformational changes, geometric constraints, atom clashes, etc.
>
> [1] Ketata M A, Laue C, Mammadov R, et al. DiffDock-PP: Rigid Protein-Protein Docking with Diffusion Models[C]//ICLR 2023-Machine Learning for Drug Discovery workshop. 2023.
>
> [2] Wang Y, Gong X, Li S, et al. xtrimoabfold: Improving antibody structure prediction without multiple sequence alignments[J]. 2022.

---

> > ### Comment · Reviewer_HJaR · 2023-08-15
> > **Response to rebuttal**
> >
> > Thank you very much for carefully responding to my questions and concerns. I personally believe that the bi-level optimization ideas provided in the paper constitute an interesting novel technical idea and perspective (which as I mentioned I think should be the emphasis of the paper). Therefore, even after reading the concerns of the other reviewers, I have raised my score to 7 and recommend acceptance.

---

> > > ### Author Response · Authors · 2023-08-15
> > >
> > > We are truly pleased to have been able to provide satisfactory explanations for the concerns you raised. Your time and attention to our rebuttal are greatly valued, and we are grateful for your willingness to adjust the score accordingly.
> > >
> > > We will diligently work on polishing the presentation and incorporating the necessary details in the rebuttal into our paper. Your support is invaluable in this process.

---

### Official Review · Reviewer_XfrH · 2023-07-05

**Soundness:** 2 fair
**Presentation:** 2 fair
**Contribution:** 2 fair
**Rating:** 4
**Confidence:** 5

**Summary:**

This paper studies the rigid protein-protein docking problem. The authors fuse the sequence features and structural features of proteins and unify these features into single features and pair features as done in AlphaFold2. These features are then fed into an evoformer-like cross-modal transformer to produce updated representations. The updated representations are used to predict inter-chain distance map and masked MSA tokens. The final rigid-docking results (i.e., a global translation and a global rotation) can be obtained by solving an optimization problem that minimizes the discrepancy between the distance map of the predicted complex and that of the ground-truth.

The authors claim that they solve the above optimization problem better by providing a better initial rotation and translation with the help of spectral initialization.

**Strengths:**

1. The proposed cross-modal transformer is clear and easy to follow.
2. The spectral initialization of the optimization problem offers good insights for those who are working on the structure prediction problems.

**Weaknesses:**

1. Some important related works are not discussed and compared. For example, (1) Diffdock-PP (https://arxiv.org/pdf/2304.03889.pdf) use torsional diffusions to solve rigid protein docking, and the source code is released. (2) McPartlon & Xu proposed DockGPT for
flexible and site-specific protein docking and design. Although they target the flexible docking, they also use evoformer to predict the inter-chain distance map. (3) Very recently, Chu, Lee-Shin, et al. propose geodock (https://www.biorxiv.org/content/10.1101/2023.06.29.547134v1),  a multi-track iterative transformer network for fast and flexible
88 protein-protein docking. (4) Finally, Luo et al. proposed xTrimoDock (https://www.biorxiv.org/content/10.1101/2023.02.06.527251v1.full.pdf), which is quite related to this work, as they also predict the inter-chain distance map and compute a global rotation and translation that best-fit the predicted distance map.
As a submission of the NeurIPS conference, I highly recommend the authors to discuss the connection of this work with the works mentioned above.

2. Figure 1. of the current manuscript is borrowed from xTrimoDock (https://www.biorxiv.org/content/10.1101/2023.02.06.527251v1.full.pdf), this could potentially be problematic.

3. The experimental results are not that clear and convincing. (1) First, Diffdock-PP is not included as a baseline. (2) Second, previous works use ligand-RMSD, complex-RMSD, or interface-RMSD to evaluate the performance of the model. How RMSD is calculated in this work is not revealed.

4. As shown in Eq. 12, 9, 11, the definition of the L^{out} loss does not involve $R^{\star}$ and $t^{\star}$. Therefore, the first term in Eq. 13 is equal to zero, and it seems that the outer loss can be directly optimized by minimizing the distance prediction loss and MSA loss. I'm not sure how the story of bilevel-optimization holds in this situation. Can the authors further clarify this point?

5. The authors claim that they use bilevel optimization to solve the problem. However, the effectiveness of the bilevel optimization is not justified.  The authors are encouraged to report the results of the variant that does not rely on bi-level optimization, i.e., directly training a distance-map predictor, and recovery the complex structures from predicted distance map.

**Questions:**

Please see the Weaknesses.

**Limitations:**

Please see the Weaknesses.

---

> ### Author Rebuttal · Authors · 2023-08-09
>
> - We sincerely thank the Reviewer for your careful reading. We would like to address the concerns by providing responses as well as additional experimental results.
>
> 1. > Some important related works are not discussed and compared. For example, Geodock, DockGPT, xTrimoDock and Diffdock-PP.
>
> **Answer:** Thanks for the suggestion. (1) GeoDock [1] and DockGPT [2] target flexible docking. Firstly, flexible docking contradicts the rigid-body assumption [3,4], rendering it incomparable with rigid docking models. Secondly, GeoDock is released after the submission deadline, explaining its absence in the references. (2) Regarding xTrimoDock, NeurIPS permits sharing of early versions on platforms like Biorxiv. (3) Before the submission deadline, issues (https://github.com/ketatam/DiffDock-PP/issues/10) with the released code of DiffDock-PP [5] exist. The authors have resolved them now, so we evaluate the performance of DiffDock-PP shown in the tables below.
>
> **Table** Quantitative comparisons between DiffDock-PP and the proposed BiDock on DB5.5 test set. (bold: best)
> ||RMSD $\downarrow$|TM-score $\uparrow$|DockQ $\uparrow$|
> | ------------------- | ----- | -------- | ------- |
> |DiffDock-PP|17.364 $\pm$ 7.262|0.670 $\pm$ 0.118 | 0.031 $\pm$ 0.047|
> |BiDock|**7.280** $\pm$ 8.117|**0.847** $\pm$ 0.158|**0.564** $\pm$ 0.369|
>
> **Table** Quantitative comparisons between DiffDock-PP and the proposed BiDock on VH-VL test set. (bold: best)
> ||RMSD $\downarrow$|TM-score $\uparrow$|DockQ $\uparrow$|
> | ------------------- | ----- | -------- | ------- |
> |DiffDock-PP|9.680 $\pm$ 5.266|0.653 $\pm$ 0.123|0.130 $\pm$ 0.115|
> |BiDock|**1.242** $\pm$ 0.602|**0.966** $\pm$ 0.021|**0.773** $\pm$ 0.187|
>
> **Table** Quantitative comparisons between DiffDock-PP and the proposed BiDock on AB-AG test set. (bold: best)
> ||RMSD $\downarrow$|TM-score $\uparrow$|DockQ $\uparrow$|maxDockQ $\uparrow$|
> | ------------------- | ----- | -------- | ------- | ----- |
> |DiffDock-PP|20.562 $\pm$ 4.237|0.578 $\pm$ 0.072|0.197 $\pm$ 0.036| 0.021 $\pm$ 0.020|
> |BiDock|**9.707** $\pm$ 8.759|**0.773** $\pm$ 0.187|**0.342** $\pm$ 0.351|**0.414** $\pm$ 0.386|
>
> From the results, BiDock generally outperforms DiffDock-PP. One possible reason is that DiffDock-PP directly utilizes MSA embeddings obtained from a pre-trained protein language model. Therefore, the learning of MSA embeddings lacks guidance from rigid structures, and PLMs trained on single chains cannot handle the pairwise contact prediction essential for docking. In contrast, BiDock employs raw MSA features and effectively leverages the rich evolutionary information through a cross-modal transformer, seamlessly integrating it with the structure-modal information.
>
> In the revised version, we will include these methods in the "Related Work" section, highlighting their differences and reporting additional results in the experiments.
>
> 2. > The experimental results are not that clear. Previous works use ligand-RMSD, complex-RMSD, or interface-RMSD to evaluate the performance of the model. How RMSD is calculated in this work is not revealed.
>
> **Answer:** Apologies for the confusion. We used complex-RMSD in our experiments. We will clarify this point in the revised version.
>
> 3. > As shown in Eq. (12), (9), (11), the definition of the $L^{out}$ loss does not involve $R^*$ and $t^*$. Therefore, the first term in Eq. (13) is equal to zero, and it seems that the outer loss can be directly optimized by minimizing the distance prediction loss and MSA loss.
>
> **Answer:** Apologies for the confusion. The outer and inner loops share the common goal of learning the optimal roto-translation transformation for rigid docking. Therefore, the inner loss $L^{in}$ in Eq. (1) is also the objective function of the outer loop. To clarify this, we will explicitly include the inner loss $L^{in}$ in Eq. (12) as:
>
> $L^{out}=\lambda_1 L_{dist}+\lambda_2 L_{msa}+\lambda_3 L^{in}$
>
> Thanks for your careful review, and we will fix this error in the revised version.
>
> 4. > The effectiveness of the bi-level optimization is not justified.
>
> **Answer:** Thanks for the suggestion. We conduct an ablation study using two-stage optimization: training outer and inner loops separately. Specifically, we use cross-entropy and masked MSA losses (Eq. (9) and Eq. (11)) to train cross-modal transformer. With the resulting distance map, we compute roto-translation transformation using inner loss and spectral initialization. The results on antibody-antigen docking are presented in the table below, with "w/o Bi" denoting the variant without bi-level optimization.
>
> **Table**	Ablation studies on bi-level optimization. (bold: best)
> ||RMSD $\downarrow$|TM-score $\uparrow$|DockQ $\uparrow$|maxDockQ $\uparrow$|
> | ------------------- | ----- | -------- | ------- | ----- |
> | BiDock w/o Bi |10.090 $\pm$ 7.817|0.733 $\pm$ 0.173|0.220 $\pm$ 0.232|0.307 $\pm$ 0.365|
> |BiDock|**9.707** $\pm$ 8.759|**0.773** $\pm$ 0.187|**0.342** $\pm$ 0.351|**0.414** $\pm$ 0.386|
>
> Results support our contributions of using bi-level optimization for end-to-end optimization, where the parameter learning of the cross-modal transformer is tailored for rigid docking.
>
> In the revised version, we will incorporate these ablation studies. Your valuable feedback is highly appreciated.
>
> [1] Chu L S, Ruffolo J A, Harmalkar A, et al. Flexible Protein-Protein Docking with a Multi-Track Iterative Transformer[J]. bioRxiv, 2023: 2023.06. 29.547134.
>
> [2] McPartlon M, Xu J. Deep Learning for Flexible and Site-Specific Protein Docking and Design[J]. bioRxiv, 2023: 2023.04. 01.535079.
>
> [3] Vakser I A. Protein-protein docking: From interaction to interactome[J]. Biophysical journal, 2014.
>
> [4] Desta I T, Porter K A, Xia B, et al. Performance and its limits in rigid body protein-protein docking[J]. Structure, 2020.
>
> [5] Ketata M A, Laue C, Mammadov R, et al. DiffDock-PP: Rigid Protein-Protein Docking with Diffusion Models[C]//ICLR Machine Learning for Drug Discovery workshop. 2023.

---

> > ### Comment · Reviewer_XfrH · 2023-08-19
> > **Thanks for authors' detailed response.**
> >
> > I have read the authors' responses as well as other reviewers' comments and I really appreciate the authors' efforts to address my questions.
> >
> > I now agree with the authors that the comparison with GeoDock and xTrimoDock is not necessary currently.
> > Nevertheless, my biggest concern about the paper, i.e., the formulation of the bilevel optimization, still remains unsolved (We note that Reviewer HJaR also mentions this issue).
> >
> > The authors mention in the replies that the inner loss in Eq. (1) is also the objective function of the outer loop (L_out = L_dist + L_msa + L_in). Such a formulation does not fit the definition of bilevel optimization. Directly adding the loss of the inner loop into the loss of the outer loop does not make much sense. (If this is the case, why don't we just simply optimize the whole outer loop and ignore anything about the inner loop.)
> >
> > The bilevel optimization is a two-step optimization process where one optimization problem (inner loop) is nested within another(outer loop). The solution of the outer loop (in this case, the learned Cross-modal Transformer which predicts the distance map) directly affects the solution of the inner loop (the optimal transformation given fixed distance map). Conversely, the solutions of the inner loop (optimal transformation given fixed distance map) should affect the optimal solution of the outer loop. As also mentioned by Reviewer HJaR, neither L_dist nor L_msa seem to depend on R nor t, which means that the solution of the inner loop does not affect the outer loop. Even if you add the loss of the inner loop into the loss of the outer loop, anything happening in the inner loop won't affect the learning of the cross-model transformation, unless you define some loss in the outer loop based on the solution of the inner loop.
> >
> > To make my point even more clear, let's imagine a hypothetical scenario. Given the solution of the inner loop (R, t), you can calculate the final docked pose of the ligand, and the whole process is differentiable (which means you can accumulate the hyper gradient of the inner process, i.e., any loss defined on the final docking pose is differentiable to the parameters of the outer loop). Let's say you define some structural loss in the outer loop (e.g., RMSD between the docked ligand pose, and the ground truth ligand pose), then the outer loop will depend on R and t, and this problem can be considered as a bi-level optimization problem. Note that you should keep track of all hyper gradients of the inner loop so that you can update the parameters of the cross-model transformer.
> >
> > I'm open to any further discussion in terms of the bilevel optimization formation of this work. I personally believe the whole story of the bilevel optimization in this work is a little bit misleading for the community and thus vote for the rejection.
> > If the authors can further clarify this point and truly fit the methodology into the framework of bilevel optimization, I'm happy to change my opinion.

---

> > > ### Author Response · Authors · 2023-08-19
> > >
> > > We value your attention to our rebuttal and thoughtful considerations, and we apologize for any misunderstandings. Let's revisit the proposed bi-level optimization framework:
> > >
> > > (1) The outer loop optimizes the cross-modal transformer to integrate multi-modal information for distance map prediction. Its loss functions comprise three parts: distogram loss (Eq. (9)) and masked MSA loss (Eq. (11)), unrelated to the inner loop, and the inner loss, associated with inner loop optimization.
> > >
> > > (2) Given the predicted distance map $\hat{D}$, the inner loop optimizes the roto-translation transformation $(R,t)$, as in Eq. (1). With optimized transformation $(\hat{R}(\phi),\hat{\mathbf{t}}(\phi))$, inner loss replaces the predicted distance map in Eq.(1) with the ground truth:
> > >
> > > $\mathcal{L}^{in}(\hat{R},\hat{\mathbf{t}},\phi)=\min_{\phi} \frac{1}{mn}\sum_{i=1}^m \sum_{j=1}^n\left(\left|X_i-\hat{R}(\phi) Y_j-\hat{\mathbf{t}}(\phi)\right|-D_{ij}\right)^2,$
> > >
> > > The misunderstanding may stem from this replacement. To clarify, our revised version will distinguish the formulation and loss of the inner loop explicitly. Notably, the optimization of roto-transformation relies on the predicted distance map, allowing hypergradients to flow through the cross-modal transformer.
> > >
> > > Furthermore, we elaborate on two points:
> > >
> > > - Why not solely optimize the cross-modal transformer using MSE between predicted and ground truth distance maps and eliminate the inner loop? The inner loop ensures rigidity.
> > > - Why not replace the predicted distance map with ground truth in the inner loop, removing the outer loop? Inference lacks ground truth, making accurate distance map prediction critical.
> > >
> > > In conclusion, these explanations illuminate interdependence and mutual enhancement of outer and inner loops in our bi-level optimization framework. We appreciate your feedback and aim to refine our work to minimize misunderstandings.

---

> > > > ### Comment · Reviewer_XfrH · 2023-08-20
> > > > **Thanks for the reply.**
> > > >
> > > > The formulation of bilevel optimization is much clearer to me now. Please update your manuscript accordingly in your next revision to make it clear. Based on the authors' replies, I think my concern about bilevel optimization has been largely addressed.
> > > >
> > > > Overall, the task this paper studied and the provided methodology are worth the audience of the ICML. However, given that the authors can not release the source code of this work and the current formulation is quite different from the previous one (I think the discrepancy is significant. Note that there is an extra term in the loss function of the outer loop now, which results in a totally different story), I have improved my score and currently lean towards to borderline reject.

---

> > > > > ### Author Response · Authors · 2023-08-20
> > > > >
> > > > > We highly value your time and attention to our rebuttal and genuinely appreciate your willingness to adjust the score based on our response.
> > > > >
> > > > > Considering the commercial nature of our proposed model, releasing the complete code presents significant challenges. Nevertheless, we are steadfast in our commitment to resolving this issue and contributing to the AI4Science open-source community. One avenue we are actively exploring involves the release of an inference interface similar to that of AlphaFold2.
> > > > >
> > > > > In this context, we also sincerely hope that you have confidence in the bi-level optimization framework we have implemented. We feel sorry that the misunderstanding surrounding one equation has led to a loss of trust. Specifically, this misunderstanding caused by not explicitly incorporating the inner loss into the outer loop in Eq. (12) can be dispelled through several key aspects: (1) If the misunderstanding holds, the model doesn't access ground truth after obtaining the final docked pose during training. This is an unreasonable scenario. (2) The comprehensive presentation of the entire model, including Fig. 2. (3) Theoretical derivations in Sect. 3.2 to solve this bi-level optimization problem. (4) Results from ablation studies, including those for Q4 in rebuttal. (5) Noteworthy performance improvements over state-of-the-art AlphaFold-Multimer.
> > > > >
> > > > > Furthermore, we want to express our gratitude for your review and thoughtful consideration, which aids in refining our presentation. In the revised version, we will meticulously polish the presentation and integrate the necessary details in the rebuttal into our paper. Your support is invaluable throughout this iterative process.

---

### Official Review · Reviewer_ZMi5 · 2023-07-05

**Soundness:** 2 fair
**Presentation:** 3 good
**Contribution:** 2 fair
**Rating:** 5
**Confidence:** 5

**Summary:**

This paper introduces BiDock, a novel approach that integrates sequence- and structure-modal information to improve the accuracy of rigid protein docking predictions. The proposed method uses multimodal information through bi-level optimization, enabling joint optimization of the docking score and the weights of the multimodal features. A gradient descent method is employed to predict the rotation and translation instead of relying on the Kabsch algorithm. Additionally, a spectral initialization is proposed for the gradient descent.
The authors demonstrate the effectiveness of BiDock by comparing it to existing docking methods on various datasets, including two newly curated datasets from PDB database. Remarkably, BiDock outperforms other methods, particularly excelling in the challenging antibody-antigen docking task.

The paper's contributions include the development of a novel approach for integrating multimodal information in protein docking, the introduction of two curated datasets, a thorough evaluation of the proposed method with baselines across multiple datasets.

**Strengths:**

- **Integration of multimodal protein representation**: The paper introduces a novel approach that combines both information representations and structural information, resulting in a comprehensive protein representation. This integration enhances the accuracy of rigid protein docking predictions.
- **Bi-level optimization formulation of the protein docking problem**: The authors formulate the docking problem as a bi-level optimization task, which allows for joint optimization of the rigid docking poses and the weights of the multimodal features. By employing a gradient descent method for predicting rotation and translation, the paper offers an alternative to the traditional Kabsch matching algorithm.
- **Spectral initialization for gradient descent**: The paper proposes a spectral initialization technique for the gradient descent method used in the bi-level optimization.
- **Introduction of two new curated datasets**: The authors curate two new datasets from the Protein Data Bank (PDB) database, specifically focusing on antibodies (VH-VL) and antibody-antigen complexes (AB-AG). These datasets provide valuable resources for evaluating protein docking methods, particularly in the context of antibody-antigen interactions.
- **Strong experimental results**: The paper demonstrates the superiority of the BiDock model over existing docking methods through comprehensive experiments on multiple datasets. The proposed method particularly excels in the challenging antibody-antigen docking task, showcasing its efficacy in accurately predicting protein docking conformations.
- **Clear presentation**: The authors provide clear explanations of the proposed approach, experimental setup, and evaluation metrics, enhancing the clarity and accessibility of the research findings.

**Weaknesses:**

- **Unclear motivation**: The paper does not clearly explain why sequential/coevolution representations are useful specifically for rigid docking problems, leaving the reader questioning the relevance and benefits of these representations in this context. Ablation study between implementation with and without sequential information is appreciated.
- **Choice of bi-level optimization**: The paper formulates the problem as a bi-level optimization task, but it lacks a clear explanation as to why this approach is chosen over direct end-to-end optimization using the distance map as input. The division of the optimization into an outer loop (distance map prediction) and inner loop (pose optimization) raises questions about the necessity and effectiveness of this bi-level optimization scheme.
- **Curated dataset filtering process**: The paper lacks detailed information about the filtering process used to curate the datasets, leaving uncertainty about how the quality of complex structures in the datasets is ensured. Without transparent and rigorous filtering criteria, the reliability of the curated datasets may be compromised.
- **Computational complexity**: BiDock exhibits higher inference time compared to existing methods like EquiDock and AlphaFold-Multimer. The computational complexity associated with the bi-level optimization process might contribute to this increased inference time, which can impact practical usability and scalability.
- **Limited training and testing data**: The paper acknowledges that the available training and testing data are limited, which raises concerns about the generalizability and robustness of the proposed method. Insufficient data may lead to overfitting and an incomplete evaluation of BiDock's performance.
- **Incremental novelty**: The paper lacks significant innovation in terms of network structure for BiDock. Instead, it builds upon existing techniques such as cross-modal transformers and roto-translation transformations, which may limit its originality and novelty compared to other state-of-the-art approaches in the field.

**Questions:**

Please see the Weaknesses part.

**Limitations:**

Yes, the authors discuss about limitation of the proposed method.

---

> ### Author Rebuttal · Authors · 2023-08-09
>
> - We highly appreciate constructive comments from the Reviewer on our work.
>
> 1. > (1) The paper does not clearly explain why sequential/coevolution representations are useful specifically for rigid docking problems. (2) Ablation study between implementation with and without sequential information is appreciated.
>
> **Answer:** Apologies for any confusion. (1) MSAs capture the evolutionary conservation of interacting residues, providing insights into residue interactions and spatial proximity. This aids in identifying crucial residues at the protein-protein interaction interface and enhances docking accuracy. (2) We perform an ablation study by retraining BiDock with all MSA features set to zero. The results for antibody-antigen docking are shown in the table below, labeled as "w/o MSA" for the variant without MSAs.
>
> **Table** Ablation studies on sequential information from MSAs. (bold: best)
> ||RMSD $\downarrow$|TM-score $\uparrow$|DockQ $\uparrow$|maxDockQ $\uparrow$|
> | ------------------- | ----- | -------- | ------- | ----- |
> | BiDock w/o MSA |20.658 $\pm$ 5.141|0.581 $\pm$ 0.094|0.0392 $\pm$ 0.081|0.046 $\pm$ 0.104|
> |BiDock|**9.707** $\pm$ 8.759|**0.773** $\pm$ 0.187|**0.342** $\pm$ 0.351|**0.414** $\pm$ 0.386|
>
> The results indicate the essential role of sequential information from MSAs in enhancing docking performance. This underscores the significance of our framework, which combines multimodal fusion and rigid body docking via bi-level optimization.
>
> In the revised version, we will incorporate these explanations and ablation studies to further bolster our contributions. Your valuable feedback is greatly appreciated.
>
> 2. > The paper formulates the problem as a bi-level optimization task, but it lacks a clear explanation as to why this approach is chosen over direct end-to-end optimization using the distance map as input.
>
> **Answer:** Thanks for your comments. The "direct end-to-end optimization" is equivalent to performing only one gradient descent step in the inner loop, approximating $(R^*, t^*)$ with $(R_1, t_1)$ in Eq. (14). Table 5 demonstrates that performance decreases when using gradient descent in the inner loop for 1000 steps compared to 2000 steps, let alone performing only one step. The reduction in gradient descent steps leads to larger approximation errors. Given this, we naturally opt for bi-level optimization over "direct end-to-end optimization."
>
> We value this question and will further emphasize the conclusions shown by the ablation studies in the revised version.
>
> 3. > The paper lacks detailed information about the filtering process used to curate the datasets, leaving uncertainty about how the quality of complex structures in the datasets is ensured.
>
> **Answer:** Thanks for your comments. The dataset extraction principles and PDB identifiers are detailed in Appendix A. Roughly speaking, the training set comprises 4,890 complexes of antibody-antigen pairs with proteins containing a minimum of 30 residues. These complexes involve three chains, including the light and heavy chains of the antibody and one antigen chain, released before January 2022. Similarly, the test set includes 68 antibody-antigen complexes with three chains, released after October 2022. This ensures no data leakage for baselines or our proposed model.
>
> Additionally, our dataset has been selected as a benchmark for the antibody-antigen complex structure prediction task on the "Life Science Leaderboard" (https://www.biomap.com/sota/), facilitating public access and fair comparisons.
>
> 4. > BiDock exhibits higher inference time compared to existing methods like EquiDock and AlphaFold-Multimer.
>
> **Answer:** Thanks for your comments. Although EquiDock is fast, its performance falls short of traditional software due to its limitations in leveraging coevolution information and simple networks. On the other hand, AlphaFold-Multimer and our proposed BiDock exhibit comparable inference times. Considering the performance improvement of our model, this trade-off is acceptable.
>
> Thanks for your thorough review. We will place greater emphasis on this point in the revised version.
>
> 5. > The paper acknowledges that the available training and testing data are limited, which raises concerns about the generalizability and robustness of the proposed method.
>
> **Answer:** Thanks for your comments. (1) The three test sets are held-out, and both our proposed BiDock and the baselines do not select checkpoints based on the evaluations of test sets. (2) Similar to the evaluation of AlphaFold2 and AlphaFold-Multimer, we save only one checkpoint after convergence and evaluate its performance on all test sets. Notably, the three test sets have distinct characteristics: DB5.5 includes general proteins; VH-VL focuses on light and heavy chains of antibodies; AB-AG concerns antibody-antigen docking. The performance improvement across these diverse test sets reflects the generalization and robustness of BiDock to some extent.
>
> We will include these clarifications in the revised version as suggested. Thanks for your feedback.
>
> 6. > The paper lacks significant innovation in terms of network structure for BiDock. Instead, it builds upon existing techniques such as cross-modal transformers and roto-translation transformations.
>
> **Answer:** Thanks for your comments. (1) Roto-translation transformation is a standard operation for 3D spatial rotation and translation, and Evoformer is a well-established module like Resnet. (2) Our contributions do not center on network structure design, emphasizing these unique aspects:
> - We design a framework to naturally integrate the fusion of multimodalities and the docking of rigid bodies through bi-level optimization establishing a new avenue for rigid protein docking.
> - We solve the above bi-level optimization with unrolled gradient and the derived spectral initialization.
> - The promising results on three representative datasets demonstrate the effectiveness of the proposed model.

---

> > ### Comment · Reviewer_ZMi5 · 2023-08-21
> > **Thank you for your response**
> >
> > Thank you for your detailed reply! I appreciate the extra analysis provided, which has indeed alleviated my concern. Nevertheless, I believe the bi-level optimization approach appears straightforward and lacks depth, leading to significant computational complexity. Overall I think the novelty of the proposed method is limited, but the task formulation might provide insight to the AI4Science community. As a result, I have adjusted my rating to a borderline acceptance level.

---

> > > ### Author Response · Authors · 2023-08-21
> > >
> > > We extend our sincere gratitude for acknowledging our rebuttal! We are genuinely delighted to have been able to provide satisfactory explanations for the concerns you raised. Your insights into our work hold significant value, and we are thankful for your readiness to adjust the score accordingly.
> > >
> > > AI4Science offers vast exploration opportunities. Given the practical nature of this field, we place equal emphasis on model novelty and performance improvements. While our proposed BiDock may not be structurally complex, it marks a good starting point regarding its novel framework and promising performance.
> > >
> > > In the revised version, we will diligently refine the presentation and incorporate the necessary details from the rebuttal into our paper. Your support is invaluable throughout this process.

---

### Official Review · Reviewer_DgVe · 2023-07-28

**Soundness:** 2 fair
**Presentation:** 2 fair
**Contribution:** 3 good
**Rating:** 5
**Confidence:** 5

**Summary:**

This paper proposes BiDock, a novel rigid protein docking model that integrates sequence and structure information through bi-level optimization. It achieves promising results, outperforming baselines by up to 234% in challenging antibody-antigen docking.

**Strengths:**

As claimed by the authors in the paper, this work represents the first attempt to effectively leverage sequence- and structure-modal information for rigid protein docking.

**Weaknesses:**

The authors seem to overlook the discussion and comparison of some protein docking methods based on pre-trained models. For example, [1] utilizes the protein language model ESM2 [2] for Protein Docking task, and there are other pre-trained models focusing on protein structures [3] as well as embeddings that combine both sequence and structure information [4]. These methods are worth discussing and applying in the experimental section for a comprehensive evaluation and comparison. In general, applying pre-trained methods directly to downstream protein-related tasks often leads to significant improvements in performance.

The paper lacks sufficient details, such as the specific approach for concatenating MSA. It is unclear whether the MSA is directly concatenated or arranged based on homology, similar to AF2-multimer. Providing these implementation details, and making the code available, would enhance the reproducibility and transparency of the research.

​​For antibody-antigen tasks, reporting the RMSDs of the CDRs are crucial. The high variability of CDRs makes predicting this region particularly challenging. Especially, results on CDR3 region is a critical metric for evaluating the model's performance, and its absence in the author's discussion is noteworthy.


[1] Chu, Lee-Shin, et al. "Flexible Protein-Protein Docking with a Multi-Track Iterative Transformer." bioRxiv (2023): 2023-06.

[2] Lin, Zeming, et al. "Evolutionary-scale prediction of atomic-level protein structure with a language model." Science 379.6637 (2023): 1123-1130.

[3] Guo, Yuzhi, et al. "Self-supervised pre-training for protein embeddings using tertiary structures." Proceedings of the AAAI Conference on Artificial Intelligence. Vol. 36. No. 6. 2022.

[4] Chen, Can, et al. "Structure-aware protein self-supervised learning." Bioinformatics 39.4 (2023): btad189.

**Questions:**

Does the proposed multi-model model have the capability to handle tasks when there is missing structure? Are there any potential improvements or modifications that could address this issue?

Do MSAs of the antibody part really work for this task?

**Limitations:**

The paper's limitations include the absence of explicit geometric constraints between residues in learning the distance map and neglecting potential atom clashes. The authors plan to address these issues and extend the framework to encompass general proteins in future work.

---

> ### Author Rebuttal · Authors · 2023-08-09
>
> - We sincerely thank the Reviewer for spending time and providing valuable feedback. We appreciate all of your suggestions and we have addressed all your questions below by providing our responses.
>
> 1. > The authors seem to overlook the discussion and comparison of some protein docking methods based on pre-trained models. For example, (1) some pre-trained models focus on protein structures [1] as well as embeddings that combine both sequence and structure information [2]. (2) GeoDock[3] utilizes the protein language model ESM2 [4] for protein docking task.
>
> **Answer:** Thanks for your comments. (1) General pre-trained models [1,2] are evaluated on simpler tasks such as classification or GraphQA. However, rigid docking requires a specialized decoder and standardized evaluation criteria is lacking. Following existing rigid docking models [7,8], we exclude general pre-trained models from our baselines. (2) GeoDock [3] is tailored for flexible docking. Firstly, flexible docking contradicts the rigid-body assumption [5,6], rendering it incomparable with rigid docking models. Secondly, GeoDock is released after the submission deadline, explaining its absence in our references.
>
> In the revised version, we will cite and discuss these methods in the "Related Work" section to differentiate the focus.
>
> 2. > The paper lacks sufficient details, such as the specific approach for concatenating MSA. It is unclear whether the MSA is directly concatenated or arranged based on homology, similar to AF2-multimer.
>
> **Answer:** Apologies for the confusion. (1) The extraction of cluster MSA features $F^{msa}\in\mathbb{R}^{N_{cls}\times N_{res}\times 49}$ is detailed in Appendix B. Roughly, we employ the heuristic approach from AlphaFold-Multimer to pair sequences, then utilize the MSA clustering method like AlphaFold2. (2) As mentioned in the paper,  Eq. (6) requires broadcasting operations. Specifically, ${M^{typ}\in\mathbb{R}^{N_{res}\times c_m}}$ is broadcasted along the newly added first dimension during addition. Similarly, broadcasting is applied to the newly added first dimension of $M^{ang}$ for concatenation.
>
> Thanks for your careful review. We will provide clear explanations in the revised version.
>
> 3. > For antibody-antigen tasks, reporting the RMSDs of the CDRs are crucial.
>
> **Answer:** Thanks for your comments. Rigid docking [5,6] ensures that the conformation of the antibody is rigid, resulting in identical CDRs across all methods. Therefore, a comparison in this regard becomes unnecessary.
>
> 4. > Does the proposed multi-model model have the capability to handle tasks when there is missing structure?
>
> **Answer:** Thanks for your comments. (1) Rigid docking [5,6] assumes access to 3D structures of unbound proteins. (2) When only amino acid sequences are available, advanced folding models like AlphaFold2 can predict the 3D structures of unbound proteins.
>
> 5. > Do MSAs of the antibody part really work for this task?
>
> **Answer:** We perform an ablation study by retraining BiDock with all MSA features set to zero. The results for antibody-antigen docking are shown in the table below, labeled as "w/o MSA" for the variant without MSAs.
>
> **Table**	Ablation studies on sequential information from MSAs. (bold: best)
> ||RMSD $\downarrow$|TM-score $\uparrow$|DockQ $\uparrow$|maxDockQ $\uparrow$|
> | ------------------- | ----- | -------- | ------- | ----- |
> | BiDock w/o MSA |20.658 $\pm$ 5.141|0.581 $\pm$ 0.094|0.0392 $\pm$ 0.081|0.046 $\pm$ 0.104|
> |BiDock|**9.707** $\pm$ 8.759|**0.773** $\pm$ 0.187|**0.342** $\pm$ 0.351|**0.414** $\pm$ 0.386|
>
> The results indicate the essential role of sequential information from MSAs in enhancing docking performance. This underscores the significance of our framework, which combines multimodal fusion and rigid body docking via bi-level optimization.
>
> In the revised version, we will incorporate these ablation studies to further support our contributions. We value your feedback.
>
> [1] Guo, Yuzhi, et al. "Self-supervised pre-training for protein embeddings using tertiary structures." Proceedings of the AAAI Conference on Artificial Intelligence. Vol. 36. No. 6. 2022.
>
> [2] Chen, Can, et al. "Structure-aware protein self-supervised learning." Bioinformatics 39.4 (2023): btad189.
>
> [3] Chu, Lee-Shin, et al. "Flexible Protein-Protein Docking with a Multi-Track Iterative Transformer." bioRxiv (2023): 2023-06.
>
> [4] Lin, Zeming, et al. "Evolutionary-scale prediction of atomic-level protein structure with a language model." Science 379.6637 (2023): 1123-1130.
>
> [5] Vakser I A. Protein-protein docking: From interaction to interactome[J]. Biophysical journal, 2014, 107(8): 1785-1793.
>
> [6] Desta I T, Porter K A, Xia B, et al. Performance and its limits in rigid body protein-protein docking[J]. Structure, 2020, 28(9): 1071-1081. e3.
>
> [7] Yan Y, Tao H, He J, et al. The HDOCK server for integrated protein–protein docking[J]. Nature protocols, 2020, 15(5): 1829-1852.
>
> [8] Ganea O E, Huang X, Bunne C, et al. Independent SE (3)-Equivariant Models for End-to-End Rigid Protein Docking[C]//International Conference on Learning Representations. 2022.

---

> > ### Comment · Reviewer_DgVe · 2023-08-19
> >
> > Thank the authors for providing detailed explanations. My concerns have been addressed, and I have raised my score.

---

> > > ### Author Response · Authors · 2023-08-19
> > >
> > > We are delighted to have been able to provide satisfactory explanations for the questions you raised. We greatly value your time and attention to our rebuttal, and we sincerely appreciate your willingness to consider adjusting the score accordingly.
> > >
> > > We will strive to enhance the presentation and incorporate the additional experiments from the rebuttal into our paper. Your support throughout this process is truly invaluable.

---

### Decision · Program_Chairs · 2023-09-21

**Decision:**

Accept (poster)

**Comment:**

This paper focuses on rigid protein docking and puts forth a new bi-level optimization approach that combines predicting a distance matrix and optimizing the predicted roto-translation accordingly.

Though the reviewer scores were mixed, after careful analysis it was decided that the paper merits acceptance.

Below, I summarize the main concerns of the reviewers and a justification of why they did not lead to a rejection of the paper:

* There were some doubts expressed about the merit of the bi-level optimization approach and specifically about how gradient flows between the two levels. It was determined after discussion that the proposed approach is valid. Reviewers expressed diverging opinions about the significance of the idea.

* A reviewer complained about the relatively small size of the training and testing data. This is a valid criticism but seems like a characteristic of the problem and not something specific to the work.

* There was some discussion about the lack of novelty in the proposed network architecture for BiDock. However, it is the opinion of the AC that the architecture doesn't need to be different for an approach to be novel. Novelty may come from the problem formulation or the way the standard components are used - here the bilevel optimization approach.

* The complexity of the method was deemed high. Though low complexity is desirable, it's not the main priority when considering a problem as hard as docking. After the problem is solved well with a slow approach the community can focus on efficiency.

Some of the initial negative opinions of the reviewers can be attributed to an imperfect presentation of the main ideas. The authors can improve upon this based on the provided feedback for the camera-ready version.